# Oscillatory surface rheotaxis of swimming *E. coli* bacteria

Arnold J.T.M. Mathijssen [1,2], Nuris Figueroa-Morales [3,5], Gaspard Junot[3], Éric Clément [3], Anke Lindner [3] & Andreas Zöttl [2,3,4]

Bacterial contamination of biological channels, catheters or water resources is a major threat to public health, which can be amplified by the ability of bacteria to swim upstream. The mechanisms of this 'rheotaxis', the reorientation with respect to flow gradients, are still poorly understood. Here, we follow individual *E. coli* bacteria swimming at surfaces under shear flow using 3D Lagrangian tracking and fluorescent flagellar labelling. Three transitions are identified with increasing shear rate: Above a first critical shear rate, bacteria shift to swimming upstream. After a second threshold, we report the discovery of an oscillatory rheotaxis. Beyond a third transition, we further observe coexistence of rheotaxis along the positive and negative vorticity directions. A theoretical analysis explains these rheotaxis regimes and predicts the corresponding critical shear rates. Our results shed light on bacterial transport and reveal strategies for contamination prevention, rheotactic cell sorting, and microswimmer navigation in complex flow environments.

[1] Department of Bioengineering, Stanford University, 443 Via Ortega, Stanford, CA 94305, USA. [2] Rudolf Peierls Centre for Theoretical Physics, University of Oxford, Oxford1 Keble Road, OX1 3NP, UK. [3] PMMH, UMR 7636 CNRS-ESPCI-PSL Research University, Sorbonne University, University Paris Diderot, 7-9 quai Saint-Bernard, 75005 Paris, France. [4] Institute for Theoretical Physics, TU Wien, Wiedner Hauptstraße 8-10, Wien, Austria. [5] Present address: Department of Biomedical Engineering, The Pennsylvania State University, University Park, PA 16802, USA. Correspondence and requests for materials should be addressed to A.L. (email: anke.lindner@espci.fr) or to A.Zöt. (email: andreas.zoettl@tuwien.ac.at)

Swimming microorganisms must respond to flows in highly diverse and complex environments, at scales ranging from open oceans to narrow capillaries[1–3]. To succeed in such diverse conditions, microbial transport often features surprising dynamics. Microswimmers can accumulate in shear flows[4–7] or behind physical obstacles[8], exhibit oscillatory trajectories and upstream motion in Poiseuille flows[6,9,10], align resonantly in oscillatory flows[11], and feature instabilities during rapid expansion[12,13]. Some of these observations were explained individually by accounting for hydrodynamics, activity and the swimmers' complex shape[6,9,14–18]. Altogether, however, the interplay of these non-linear properties is far from trivial and remains largely unexplored.

Surfaces are crucial for bacterial transport because many species accumulate on boundaries[19–21]. Moreover, upstream swimming is facilitated by no-slip surfaces because there the counterflows are weak[18,22], but this confinement also changes their dynamics in surprising ways[23–28]. In quiescent liquids, swimmers move in circles[29–31], but in currents they can orient with respect to gradients in the flow velocity—an effect called "rheotaxis"[32]. In particular, organisms can reorient to migrate upstream, as observed for sperm cells[32–34], for *E. coli* bacteria[35–38] and artificial microswimmers[39–42]. This upstream motion has been analysed theoretically[42–45] and is generally attributed to fore-aft asymmetry of the swimmer shape. A second type of rheotaxis, at higher flow rates, can reorient organisms towards the vorticity direction[35–37,46], which is attributed to the inherent flagellar chirality[17]. Moreover, bacterial rheotaxis at surfaces has been quantified by measuring instantaneous orientation distributions[36] or average transport velocities[37], but a dynamical picture of the underlying mechanisms is still missing.

Here, we investigate, for the first time, the time-resolved orientation dynamics of *E.coli* bacteria, as a function of applied shear close to the walls. Two recent experimental techniques are combined with Brownian dynamics (BD) simulations and a theoretical analysis. With increasing flow, we identify four regimes separated by critical shear rates: (I) the well-known circular swimming; (II) direct upstream swimming without circling and without oscillations; (III) a novel oscillatory motion, biased towards the direction of positive vorticity; (IV) coexistence of oscillatory swimming to the positive and negative vorticity directions, with dynamical switching between these states. By monitoring the bacteria with 3D Lagrangian tracking we examine these regimes as a function of the shear rate. In a second assay, the bacterial flagella are stained fluorescently to reveal their cellular orientation dynamics and the oscillation frequencies. Matching these experiments, we model the bacterial rheotaxis by accounting for the cells' chiral nature, hydrodynamic and steric interactions with surfaces, elongation, fore-aft asymmetry and activity. Starting from these individual swimmer–surface-flow interactions we can fully reconstruct the observed motility, and explain how this can be applied to other bacterial species. Hence, these findings provide a broad understanding of microbial swimming in confined flows and allow to raise suggestions for optimising flow geometries as, for example, antibacterial channel design.

## Results

**Experimental observations**. We observe the dynamics of *E. coli* bacteria swimming at surfaces under flow (Fig. 1a). Two independent experimental realisations are used (see the sections from '3D tracking experiments' to 'Definition of shear rate' under Methods for experimental details). First, we employ a recently developed 3D tracking technique[47] that provides full 3D trajectories of swimming bacteria over large distances, revealing their

long-time Lagrangian dynamics. The bacteria used in these experiments are smooth swimmers, *E. coli* strain CR20, a mutant that almost never tumbles and moves with a typical swimming speed of $v_s = (26 \pm 4)\,\mu\text{m s}^{-1}$. The experimental device is a rectangular channel made in PDMS and a constant flow is applied with a syringe pump. The channel height is $H = 100\,\mu\text{m}$, the width $W = 600\,\mu\text{m}$ and its length is of several millimetres. Bacterial trajectories are only selected when they are located more than $100\,\mu\text{m}$ from the lateral side walls and $<5\,\mu\text{m}$ from the bottom surface, so that the shear rate is approximately constant, $\dot{\gamma} = 4V_{max}/H$, where $V_{max}$ is the maximum flow velocity at the channel centreline. Close to the bottom surface 3D trajectories are nearly identical to the $x$–$z$ projections.

Typical 3D trajectories for shear rates $\dot{\gamma} = 1 - 50\,\text{s}^{-1}$ are displayed in Fig. 1b, in the laboratory frame. With increasing shear, we observe a range of different dynamics. Interestingly, at small shear rates (blue trajectories) the well-known circular motion[29–31] starts to evolve towards cycloid motion with a bias "to the right". Here we define the term "to the right" as the direction of the vorticity vector, $\mathbf{\Omega}_f = -\dot{\gamma}\hat{z}$ (Fig. 1a; green arrow). This left-right symmetry breaking stems from the chirality of the bacterial flagella, as discussed below. Subsequently, at intermediate shear (cyan tracks) circles are suppressed and, instead, upstream motion is observed. When further increasing the shear rate (orange tracks) bacteria are transported downstream more strongly and the laboratory frame trajectories bend into the direction of the flow. These trajectories are mostly oriented towards the right, as reported previously[36]. However, for the first time we observe that swimming towards the left can also occur at high shear rates (red tracks). Note that different types of trajectories may coexist, as explained later, due to variations in bacterial shape, the distance from the wall and other sources of noise inherent to living bacteria.

Surprisingly, an oscillatory motion appears in these trajectories at frequencies very different from the flagellar and body rotation. These undulations are visible in the trajectories at the higher shear rates (orange and red tracks) and can be identified clearly by looking at $v_z(t)$, the velocity component transverse to the flow direction (Fig. 1c). Since the 3D tracking technique does not provide direct access to bacterial orientation, we also perform a second and complementary set of experiments (see the section 'Fluorescence experiments' under Methods). Here we use a genetically modified strain of bacteria, from the AB1157 wild-type (AD1)[48], with a fluorescently labelled body and flagella so that the cell orientation is directly visualised. This wild-type strain can tumble, but we only select trajectories without tumbles, which can be easily identified from the images. The channel dimensions are $H = 20\,\mu\text{m}$ and $W = 200\,\mu\text{m}$, and again bacterial dynamics are captured only within a maximal distance of $5\,\mu\text{m}$ from the bottom surface. In this strong confinement, high shear rates can be obtained using relatively small flow velocities, which facilitates straightforward manual tracking, but at the cost of a more variable shear rate in $y$ compared with the 3D tracking. These fluorescence experiments unambiguously demonstrate the existence of oscillatory motion around a stable position (Fig. 1e), shown here for an example swimming to the right. Moreover, they provide an immediate measure of the orientation angle dynamics $\psi(t)$ (Fig. 1d).

To quantify this oscillatory rheotaxis, for the 3D tracking and fluorescence assays, we extract the oscillation frequencies from Fourier transformation of $v_z(t)$ and $\psi(t)$ respectively (see the section 'Data analysis' under Methods). In both experiments the measured frequencies indicate a cross-over (Fig. 1f). At small shear rates we find a constant frequency, corresponding to the circular swimming, and after a certain shear ($\dot{\gamma} \approx 15\,\text{s}^{-1}$) we observe an increase of the frequency corresponding to the

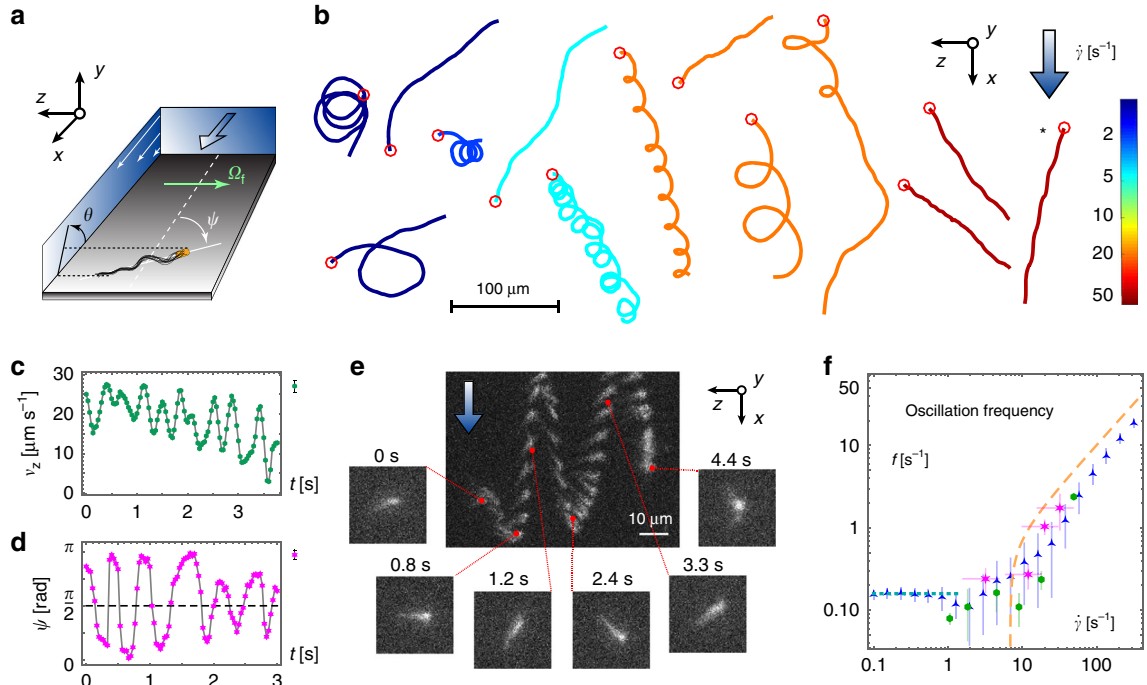

**Fig. 1** Experimental observations of oscillatory rheotaxis. **a** Set-up geometry. **b** Various types of surface trajectories obtained from 3D tracking at shear rates $\dot{\gamma} = 1 - 50\,\text{s}^{-1}$ (colours), shown in the lab frame and arranged according to increasing shear. Circles indicate the initial positions. **c–f** Oscillatory rheotaxis. **c** Typical temporal evolution of the transverse velocity $v_z(t)$ from a 3D tracking experiment that oscillates to the left (marked with an asterisk in panel **b**), at $\dot{\gamma} = 49\,\text{s}^{-1}$. The uncertainty $\Delta v_z \approx 1.5\,\mu\text{m s}^{-1}$ (error bars; top right). **d** Typical temporal evolution of the in-plane angle $\psi(t)$ from a fluorescence experiment at $\dot{\gamma} = 32\,\text{s}^{-1}$. The uncertainty $\Delta \psi \approx 5.5°$ (error bars; top right). **e** Time lapse of an oscillating bacterium with fluorescently stained flagella, using 10 fps snapshots overlaid to highlight its trajectory, taken in the Lagrangian reference frame of the average downstream bacterial velocity. **f** Oscillation frequency versus shear rate, obtained from Fourier transformation of $v_z(t)$ in 3D tracking experiments (green hexagons), of $\psi(t)$ in fluorescence experiments (magenta stars), and of $\psi(t)$ in simulations (blue triangles). Horizontal error bars stem from the uncertainty of the bacterial $z$ position, and vertical error bars correspond to two standard deviations from the ensemble mean, the 95% confidence interval. Ninety-four trajectories from 3D tracking experiments and 34 trajectories from fluorescence experiments, corresponding each to a different bacteria, have been analysed. Overlaid are theoretical estimates for the oscillation frequency (Eq. (23); dashed yellow line) and the circling frequency ($\nu_C/2\pi$; dotted blue line). Parameters used are: $\nu_W = 4\,\text{s}^{-1}$, $\theta_0 = -10°$, $\nu_C = 1\,\text{s}^{-1}$, $\Gamma = 4$, $\bar{\nu}_H = 0.02$, $\bar{\nu}_V = 0.5$, $\theta_V = 2.3°$, $\theta_E = 20°$, $v_s = 20\,\mu\text{m s}^{-1}$, $h_s = 1\,\mu\text{m}$, $D_r = 0.057\,\text{s}^{-1}$

oscillatory trajectories. Note, oscillatory rheotaxis should not be confused with the wobbling dynamics due to flagellar rotation[49], which have much higher frequencies and are distinctly different.

In the next sections we will develop a comprehensive model that explains these complex dynamics and predicts the corresponding oscillation frequencies.

**Theoretical building blocks.** In order to understand the rich behaviour of our experimental findings, we first identify and summarise the individual mechanisms that affect bacterial orientations. We distinguish between wall effects, flow effects, and the coupling of these. In the next sections we then combine these building blocks and describe their non-trivial interplay.

We model a bacterium consisting of an elongated body and a left-handed flagellar bundle, subject to shear flow at a surface (Fig. 2). We explicitly model both the in-plane angle $\psi \in \{-\pi, \pi\}$ and the pitch (i.e. dipping) angle $\theta \in \{-\pi/2, \pi/2\}$ (Fig. 1a). The bacterial conformation in principle also depends on the flagellar helix phase angle, which can lead to phase-dependent wobbling motion[49,50], but owing to its fast flagellar rotation this angle is averaged over. The orientation of a swimmer at the surface then evolves as

$$\dot{\psi} = \Omega_\psi(\psi, \theta), \qquad \dot{\theta} = \Omega_\theta(\psi, \theta), \qquad (1)$$

where the reorientation rates $\Omega_\psi$ and $\Omega_\theta$ stem from three main contributions, $\Omega = \Omega^W + \Omega^F + \Omega^V$, that account for the presence

of the wall ($\Omega^W$), local shear flow ($\Omega^F$), and surface-flow coupled effects ($\Omega^V$).

First, we discuss the wall effects. In the absence of flow, hydrodynamic swimmer-wall interactions[20,51] and steric interactions[21] enable bacteria to swim at a stable orientation approximately parallel to the wall[52–54] (Fig. 2a). We model this surface alignment as $\Omega_\theta^W(\theta) = -\nu_W \sin 2(\theta - \theta_0)$, where the prefactor $\nu_W$ is an effective angular velocity capturing both the hydrodynamic and steric contributions, and the small zero-shear pitch angle $\theta_0 < 0$ represents the observation that bacteria on average point towards the wall[49]. The second wall effect stems from the counter-rotation of the bacterial head and flagellar bundle. Near solid surfaces this leads to a hydrodynamic torque leading to circular motion in the clock-wise direction[31,51,52] (Fig. 2b). The associated reorientation rate in the $\psi$ direction is approximated by a constant, $\Omega_\psi^W(\theta) = \nu_C$. For typical bacteria with left-handed flagella the prefactor is positive, $\nu_C > 0$, giving clock-wise circles.

Second, we discuss the contributions due to shear flows, $\Omega^F = \Omega^J + \Omega^H$. Elongated objects such as rods and fibres, or dead bacteria[55], perform Jeffery orbits[56] such that the orientation vector performs a periodic motion about the vorticity ($z$) direction (Fig. 2d), given by $\Omega_\psi^J = \frac{\dot{\gamma}}{2}(1 + G)\sin\psi\tan\theta$ and $\Omega_\theta^J = \frac{\dot{\gamma}}{2}(1 - G\cos 2\theta)\cos\psi$, where $G = \frac{\Gamma^2 - 1}{\Gamma^2 + 1} \lesssim 1$ is a geometric factor describing the elongation of the bacterium with effective aspect ratio $\Gamma$. In the presence of walls, the orbit amplitudes decay

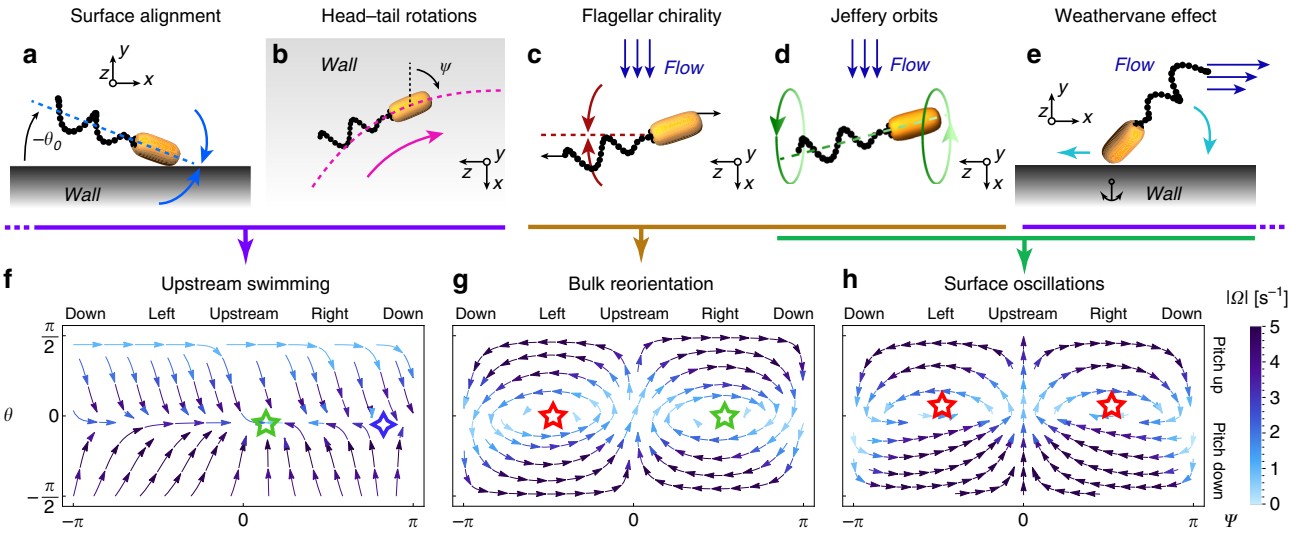

**Fig. 2** Summary of reorientation mechanisms included in the model. Wall effects: **a** Steric and hydrodynamic interactions align swimmers with surfaces. **b** Clock-wise torque from the counter-rotation of the cell body and flagella. Flow effects: **c** Left-handed helical flagellar bundle in shear reorients swimmers to the right. **d** Jeffery orbits of elongated bacteria. Flow-wall coupling: **e** Weathervane effect reorients swimmers to the upstream direction. For all these individual contributions (**a**–**e**), the corresponding orientation dynamics in $\psi$–$\theta$ phase space are shown in Supplementary Fig. 1. Combinations of the different effects give **f** swimming in the upstream direction (**a**, **b**, **e**), **g** bulk reorientation, biased to the right due to flagellar chirality (**c**, **d**), and **h** oscillatory swimming, oriented slightly upstream due to the weathervane effect (**d**, **e**). Green (red) stars are stable (unstable) orientation fixed points, and the blue diamond is a saddle point. The parameters used are given in the caption of Fig. 1, with shear rate $\dot{\gamma} = 5\,\mathrm{s}^{-1}$

because of the surface alignment but their reorientation rate (frequency) is not affected significantly, as simulated in detail for passive ellipsoidal particles[57]. The second flow effect stems from the chirality of the bacterial flagella, making cells reorient towards the vorticity direction[17,58]. Together with activity this enables stream-line crossing, which in the bulk leads to a net migration of bacteria to the right[46]. We compute this effect using resistive force theory, extending the calculations in ref. [46] for all body cell orientations (see the section 'Chirality-induced rheotaxis' under Methods and Fig. 2c). This yields the chirality-induced reorientation rates

$$\Omega_\psi^H = \dot{\gamma}\bar{\nu}_H \cos\psi \frac{\cos 2\theta}{\cos\theta}, \quad \Omega_\theta^H = \dot{\gamma}\bar{\nu}_H \sin\psi \sin\theta. \quad (2)$$

The prefactor solely depends on the geometry of the bacterium, $0 < \bar{\nu}_H \ll 1$ for left-handed flagella.

Third, we introduce the weathervane effect, a term that has been identified as an important contribution for sperm rheotaxis[33,34]. The swimmer body experiences an effective anchoring to the surface when pointing towards it, because its hydrodynamic friction with the wall is larger than that of the flagellar bundle[59], an effect explained by lubrication theory[34]. Consequently, the flagella are advected with the flow, like a weathervane (Fig. 2e). Then, the bacterium orients upstream[33,34,60], which we model using the reorientation rates

$$\Omega_\alpha^V = -\dot{\gamma}\bar{\nu}_V \sin(\alpha)\left[\frac{1}{2}\left(1 - \tanh\frac{\theta}{\theta_V}\right)\right], \quad (3)$$

for both $\alpha = \{\theta, \psi\}$. The hyperbolic tangent, with a constant $\theta_V$ depending on the cell geometry, accounts for the fact that the asymmetry in friction reduces when the swimmer faces away from the surface, $\theta > 0$, where the weathervane effect disappears. This notion was not included in previous single-angle descriptions[33,34].

**Combining the individual reorientation mechanisms.** Having described the individual effects of the wall and the flow on bacterial reorientation, we can now combine these terms to begin to understand more complex dynamics. First of all, by joining the

contributions from surface alignment and head-tail rotations, we recover the well-known circular swimming[31,52]. However, when we also add the weathervane effect Eq. (3) the cells break out of the circular kinetics and swim upstream, which corresponds to a stable fixed point in their orientation space (Fig. 2f). This Adler transition has also been observed for sperm cells[34].

Second, combining the effects of Jeffery orbits and chirality Eq. (2), we recover bulk rheotaxis[46]. Recast into the language of dynamical systems, the symmetry breaking leading to preferred motion to the right can be classified as a stable spiral fixed point in $\psi$–$\theta$ phase space (Fig. 2g).

Third, merging the Jeffery orbits in the bulk and the weathervane effect Eq. (3) for cells near a surface, we find that Jeffery's periodic motion about the vorticity directions ($\pm\hat{z}$) now shifts to oscillations about a vector pointing more and more upstream. This already hints towards the observed oscillatory rheotaxis (Fig. 2h). To understand the experimental trajectories accurately, however, we must include all terms together and also add fluctuations, as described in the next section.

**Brownian dynamics simulations.** Combining all the aforementioned contributions, we solve the orientation dynamics by integrating Eq. (1) together with rotational noise in Brownian dynamics (BD) simulations, for a broad range of applied shear rates (see the section 'BD simulations of surface rheotaxis' under Methods for simulation details). All parameter values have been estimated carefully from established experiments, simulations and theoretical arguments from the literature (see the section 'Estimation of basis parameter set' under Methods).

A simulated trajectory starts with a random in-plane angle $\psi$, parallel to the surface, $\theta = 0$, and it finishes when it reaches a given escape angle, $\theta_E$[49,61–63]. Subsequently, the spatial dynamics are found by computing the velocity parallel to the surface, at a constant swimming speed $v_s$ plus the downstream advection with velocity $\mathbf{v}_f = \dot{\gamma}h_s\hat{\mathbf{x}}$ based on the shear rate and the distance from the wall, $h_s$. Hence, Fig. 3 shows typical trajectories and the orientation dynamics in $\psi$–$\theta$ space for different shear rates $\dot{\gamma}$ and

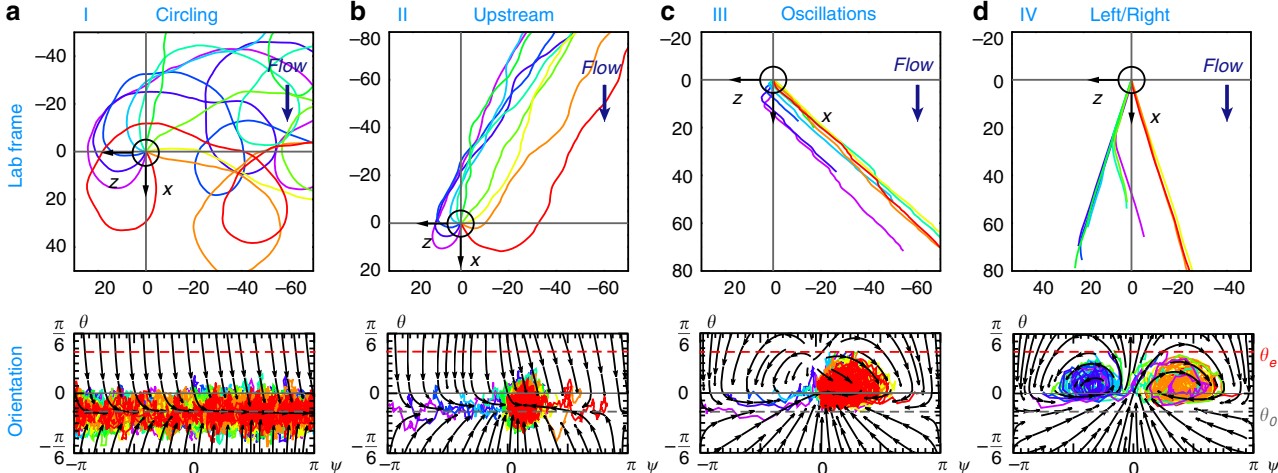

**Fig. 3** Characterisation of the four different surface rheotaxis regimes. Shown are simulated trajectories in the laboratory frame (upper panels) and the corresponding orientation dynamics (lower panels) with increasing shear rate: **a** (I, $\dot{\gamma} = 1\,\mathrm{s}^{-1}$) Circular swimming with a bias to the right. **b** (II, $\dot{\gamma} = 5\,\mathrm{s}^{-1}$) Upstream motion. **c** (III, $\dot{\gamma} = 25\,\mathrm{s}^{-1}$) Oscillatory motion, increasingly more to the right. **d** (IV, $\dot{\gamma} = 60\,\mathrm{s}^{-1}$) Coexistence between swimming to the right and to the left, with dynamical switching between these. Black circles indicate the initial swimmer positions. The parameters used are given in the caption of Fig. 1

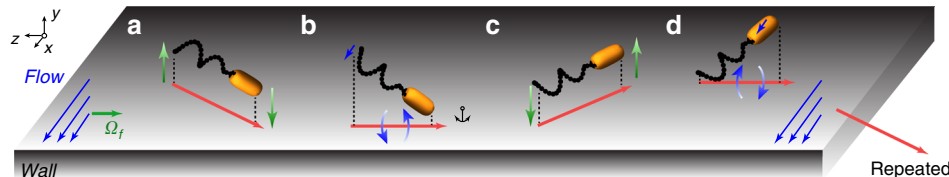

**Fig. 4** Sketch of the oscillatory rheotaxis mechanism. Here the bacterium is initially oriented towards the right and slightly downstream, and red arrows show the projection of the cell onto the surface. Then, the oscillations can be envisaged as a 4-step process: **a** The vorticity pushes the body down onto the surface and lifts the flagella up. **b** Then the flow advects the flagella faster than the body, rotating the bacterium about the $y$ axis to the upstream direction. The weathervane effect enhances this rotation as the cell pivots about the anchoring point. **c** Now the vorticity pushes the flagella onto the wall and lifts the body up. **d** Subsequently the body is advected faster, rotating the swimmer back to the downstream direction. This cycle is repeated, leading to oscillatory trajectories. Note that this is a simplified picture and all surface and flow effects (Fig. 2) contribute to the dynamics at any one time

initial conditions. We identify four regimes (I–IV) separated by critical shear rates:

At weak flows (regime I) the bacteria move in circular trajectories, with a drift to the right (Fig. 3a). Above a critical shear rate, found in the BD simulations at $\dot{\gamma}_{c_1}^{\mathrm{sim}} \approx 1.5\,\mathrm{s}^{-1}$, they no longer move in circles but swim stable to the right and slightly upstream (regime II) (Fig. 3b). This Adler transition[34] stems from the competition between the constant head-tail rotations and the weathervane effect that increases with flow. Owing to noise, coexistence between circling and upstream swimming may exist close to $\dot{\gamma}_{c_1}$, and oscillations may also appear already, as discussed below.

Above a second critical shear rate, $\dot{\gamma}_{c_2}^{\mathrm{sim}} \approx 15\,\mathrm{s}^{-1}$ (regime III), an oscillatory motion directed to the right emerges (Fig. 3c). Similar to the first transition, the oscillations arise because the flow contributions now outweigh the surface terms that do not increase with shear. Particularly the Jeffery and weathervane effects govern the oscillation dynamics, as discussed in the theoretical predictions section. A simplified pictorial summary of this oscillation process is provided in Fig. 4. However, the equilibrium angles about which the cells oscillate, $\psi^* \sim \frac{\pi}{2}$ and $\theta^* \sim 0$, still depend strongly on the other terms, as derived below. Especially the surface alignment is necessary for stability, so in general the dynamics remain a complex interplay between all contributions and fluctuations.

Above a third critical shear rate, $\dot{\gamma}_{c_3}^{\mathrm{sim}} \approx 40\,\mathrm{s}^{-1}$ (regime IV), oscillatory swimming to the left arises (Fig. 3d), in coexistence with the aforementioned oscillations to the right. In phase space, this is defined by the emergence of a stable spiral fixed point, at $\psi^* \sim -\frac{\pi}{2}$ on the left. Moreover, bacteria may switch dynamically between the left and right (purple and green trajectories). Still, this mode of leftward rheotaxis is rare as the flagellar chirality gives a bias to the right.

Throughout these regimes, the orientation distributions have a complex dependence on the shear rate (Fig. 5a, b). In the absence of flow, the in-plane orientation $\psi$ is uniformly distributed and the pitch angle $\theta$ is peaked around $\theta_0 = -10°$, as expected (dark blue distribution). In regime I the circular swimming is biased to the right, giving a peak in the distribution at $\psi^* \sim \frac{\pi}{2}$ (light blue). In regime II the swimmers move more and more upstream, so the peak shifts to $\psi^* \gtrsim 0$ (green) due to the weathervane effect, as also observed for sperm cells[34]. Meanwhile the pitch angle $\theta$ gradually increases. In regime III the cells shift from upstream to the right again, $\psi^* \sim \frac{\pi}{2}$ (yellow), in agreement with previous studies[35–37]. This is explained by the weakening of the weathervane effect, because the surface anchoring is reduced by the Jeffery term that tries to rotate the bacterium away from the surface. Consequently, there is an optimal shear rate for upstream orientation, around $\dot{\gamma}_{u}^{\mathrm{sim}} \approx 13\,\mathrm{s}^{-1}$. Finally, in regime IV swimming to the left emerges at $\psi^* \sim -\frac{\pi}{2}$ (red). At these high shear rates the pitch angle

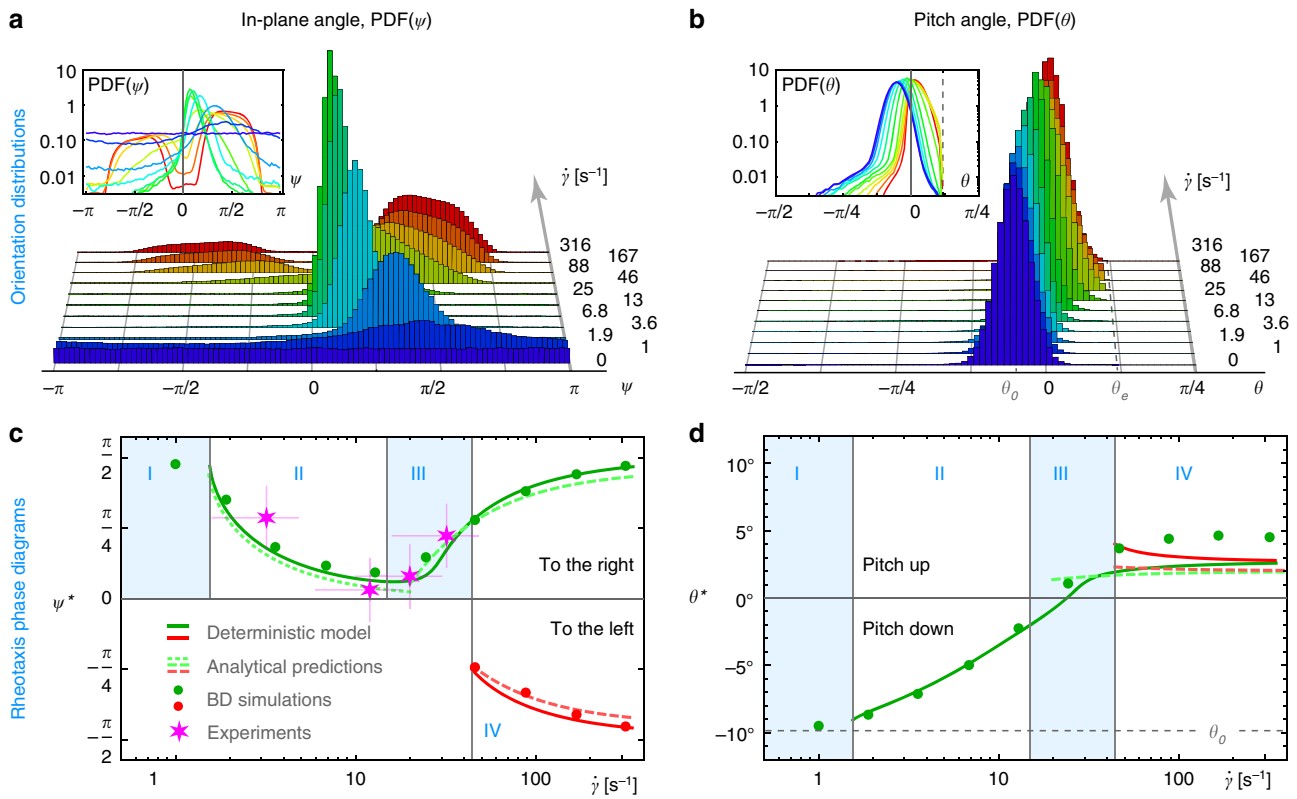

**Fig. 5** Bacterial orientation as a function of applied shear. **a**, **b** Distributions of the in-plane angle $\psi$ and the pitch angle $\theta$, respectively, obtained by simulating $N = 10^4$ trajectories over long times until steady state is reached, for different shear rates $\dot{\gamma}$. The parameters used are the same as in Fig. 1. Inset: same on a logarithmic scale. **c**, **d** Rheotaxis phase diagrams. Shown are the equilibrium in-plane angle $\psi^\star$ and the equilibrium pitch angle $\theta^\star$, respectively, obtained numerically from the full deterministic model (solid lines), analytically (dashed lines), from simulations (points) and from experiments (magenta stars). The four regimes (blue and white areas) are separated by critical shear rates $\dot{\gamma}_c$

reaches the escape value $\theta_E$ more frequently, so the residence time that bacteria spend on the surface also decreases significantly.

**Analytical predictions.** The observed and simulated transitions can also be met with theoretical estimates (see the sections from 'Summary of equations of motion' to 'Approximation of the equilibrium orientations' under Methods for theory details). We consider the case in the absence of noise, and analyse the dynamics in the $(\psi, \theta)$ orientation space. The equilibrium orientations, $\psi^\star$ and $\theta^\star$, are defined as the stable fixed points of Eq. (1). As shown in Fig. 5c, d, we first obtain their exact values numerically as a function of shear rate (solid lines), where the dynamics to the right/left are shown in green/red. Importantly, these deterministic solutions agree well with the average orientations in the BD simulations (green and red points).

Using this framework several analytical predictions are made. Near the Adler transition we find the equilibrium orientation $\psi^\star \approx \arcsin(\dot{\gamma}_{c_1}^{th}/\dot{\gamma})$ (dotted line), where the first critical shear rate is

$$\dot{\gamma}_{c_1}^{th} = \frac{2\nu_C}{\bar{\nu}_V\left(1 - \tanh\left(\frac{\theta_0}{\theta_V}\right)\right) - (1+G)\tan(\theta_0)}, \quad (4)$$

which evaluates to $\dot{\gamma}_{c_1}^{th} \approx 1.5\,\mathrm{s}^{-1}$. For larger shear rates we linearise Eq. (1) and solve for the two roots $(\psi^\star, \theta^\star) = (\varepsilon_\psi \pm \frac{\pi}{2}, \varepsilon_\theta)$ such that $\Omega = 0$, where the $\pm$ signs denote motion to the right/left (dashed green/red lines). Then, the second critical shear rate, the onset of oscillations, is defined where the eigenvalues about the right fixed point $(\psi_R^\star, \theta_R^\star)$ develop an imaginary component. We compute this numerically, which

yields $\dot{\gamma}_{c_2}^{th} \approx 24\,\mathrm{s}^{-1}$. The third critical shear rate is defined where the left fixed point $(\psi_L^\star, \theta_L^\star)$ becomes stable, which is also found numerically, $\dot{\gamma}_{c_3}^{th} \approx 44\,\mathrm{s}^{-1}$. Moreover, the optimal shear rate for being oriented upstream is found to be $\dot{\gamma}_u^{th} \approx 16\,\mathrm{s}^{-1}$. Finally, we also derive the oscillation frequency Eq. (23), shown in Fig. 1f (yellow line). For large shear rates this frequency can further be approximated as

$$\omega_O = \frac{\dot{\gamma}}{2}\sqrt{(1 - G^2) + (1 - G)\frac{\bar{\nu}_V}{\theta_V} - \frac{1}{4}(\bar{\nu}_V \pm 4\bar{\nu}_H)^2}, \quad (5)$$

which increases linearly with the applied shear rate.

These predictions offer good agreement with our BD simulations and experiments. Having said that, there are subtle differences between the deterministic model and the noisy dynamics, since fluctuations do contribute to three main effects: First, the fluctuations can sustain oscillations. In the deterministic model, the fixed points in regimes III and IV are stable spirals with eigenvalues that feature an imaginary component but also a small negative real component. Therefore, in the absence of noise, the cell orientations converge to these stable spirals, such that oscillations are damped out slowly. However, fluctuations about these points maintain finite oscillation amplitudes. Second, the noise can shift the critical shear rates. Above $\dot{\gamma}_{c_1}^{th}$ circular swimming can coexist due the stochastic Adler equation. The noise also allows for oscillations to emerge below the second critical shear rate, because it drives the system away from the stable star fixed point into an oscillatory region of phase space. Third, fluctuations facilitate dynamical switching between left and right-orientated rheotaxis (Fig. 3d), which can be envisaged as

jumps in the orientation space between two locally stable attractors. Finally, bacterial tumbling in wild-type cells further enhances noise-induced effects such as the ability to swim to the left: For example, at $\dot{\gamma} = 46\ \text{s}^{-1}$ the fraction of bacteria oriented to the left is about 11% in the absence of tumbling (Fig. 5a), but with tumbles this increases substantially, to about 22%, as dynamical switching is enhanced (Supplementary Fig. 2).

**Comparison with experiments**. We first compare the swimmer dynamics itself: All types of trajectories predicted by our model (Fig. 3) are also observed in the 3D tracking experiments (Fig. 1b). At lower shear rates (light blue trajectories) we see the transition from circular motion to upstream swimming. At intermediate shear rates (orange trajectories) we see a transition from smooth swimming to oscillations. At the highest shear rates (red trajectories) we also see the switching to left-oriented motion. Of course, in these observations visual differences can arise from fluctuations, variations in swimming speed and distance with respect to the wall. Consequently, near the thresholds these regimes coexist. The critical shear rates predicted from both numerical and analytical findings fall within these experimental coexistence intervals.

Secondly, the angular dynamics are well captured. From our fluorescence experiments we compute the peak in the distribution of in-plane angles, which are shown in the phase diagram (Fig. 5c; magenta stars). As predicted, at low shear rates the motion is mostly to the right. Then, at intermediate shear we see predominantly upstream orientations. At the highest shear rates the most common orientation is to the right again. Here a small fraction also moves to the left, around one in five bacteria, compared with ~15% in the BD simulations at the same shear and also with tumbling (Supplementary Fig. 2).

Finally, the oscillation frequencies from experiments and BD simulations are in good agreement, as shown in Fig. 1f. They all show the cross-over from a constant circling frequency (dotted blue line) to oscillations with an increased frequency at higher shear. This non-trivial oscillation frequency is described quantitatively by the analytical prediction (yellow line).

Beyond the *E. coli* bacteria studied in this paper, our model may also be applicable to a wide range of other species, as well as synthetic microswimmers. As described in the section 'Estimation of basis parameter set' under Methods, the model parameters only depend on the physical features of the swimmer, and they can be estimated directly from experimental measurements and theoretical predictions. Moreover, in the section 'Robustness in parameter variations' under Methods we show that our results are qualitatively robust against changes in these parameters, and we give a physical intuition how each term alters the bacterial dynamics. Importantly, all four rheotaxis regimes always exist for all parameters tested. The dynamics remain qualitatively the same, while the critical shear rates are only shifted in value. With this understanding for different parameters, we expect that surface rheotaxis can be applied to other swimmers in much broader contexts.

## Discussion

In this paper we investigated the full time-resolved orientation dynamics of *E. coli* bacteria close to surfaces, as a function of the applied flow strength. We demonstrated that bacterial surface rheotaxis can be categorised in four regimes, separated by shear-regulated transitions. We observe these regimes using a Lagrangian 3D cell tracking technique that allows us to follow bacteria in a flow over long times, and with independent measurements by fluorescently staining the cell flagella to monitor the cell orientation explicitly. A comprehensive model delineates these

dynamics by combining previously postulated contributions with terms derived here for rheotaxis near a surface. Simulating this model yields the cellular orientation distributions and their oscillation frequencies with increasing shear rates, and a theoretical analysis of these equations allows to predict the corresponding critical shear rates.

The bacterial contamination potential[22] is influenced by the characteristics of all these regimes. In biological and medical settings, flow rates vary across a broad range and often vary over time. For example, a typical urine catheter of inner diameter 4.4 mm with a constant flow rate of 1.5 L/day[64] is subject to a maximum shear rate of 2.9 s$^{-1}$ [36], in the upstream swimming regime (II). However, this shear can be a hundred times larger during a 20 s release[65], well within regime (IV). Another typical example in domestic pipes of diameter 4 cm gives 5.3 s$^{-1}$ for a small faucet (2 L/min) or 26.5 s$^{-1}$ for a shower (10 L/min)[66]. Therefore, to evaluate the contamination potential, it is important to understand the orientation dynamics for a broad spectrum of shear rates. It is crucial to quantify how fast the cells can move upstream in regime II, but also how well they can fight the downstream advection in regimes III–IV, because, on average, most bacteria on the surface are still oriented upstream (Fig. 5). Moreover, the orientation dynamics will also determine whether the cells will stay at the surface, or whether they will detach and be rinsed down with much larger bulk velocities. If the cells stay at the surface and reduce the downstream motion, they can still go upstream on average as the flow varies in time. Similarly, the shear rate in biological and medical systems will also vary in space, e.g. because a channel is wider and narrower in different places, so the shear is modified. Different places in the device will then correspond to different rheotaxis regimes (I–IV) that the bacteria can swim into. Subsequently, the dynamics in the local regimes (III–IV) will again affect the global upstream swimming ability.

Using this knowledge, it might be possible to invent microfluidic channel designs that prevent bacterial contamination. Building on the idea that a large majority of bacteria tend to swim with a bias towards the vorticity direction[35–37,46], one could deter upstream swimming in conventional cylindrical pipes with a right-handed surface patterning that spirals inside the duct, but without creating sharp corners that might promote upstream swimming again[37]. The helix angle of this patterning should then be tuned according to the bacterial $\psi$ dynamics. Another strategy builds on the idea that oscillations (regime III) increase the probability for bacteria to detach from the surface, so that they are subjected to a strong downstream advection in the bulk. Designing channels such that the surface shear is always larger than $\dot{\gamma}_{c_2}$ could modify the average residence time on the wall and thus the ability for cells to contaminate upstream areas or initiate biofilms. Beyond bacterial contamination one could also think of new mechanisms to separate cells, by rheotactic sorting. Differences in the geometry can significantly change the orientation dynamics, as cells with larger aspect ratios tend to swim more to the right[35]. We can understand this now, as a smaller body and longer tail gives rise to a weaker weathervane effect. New devices can thus be tuned to differentiate these geometries. One can also better understand motility in complex confinement under flow, like pathogens in wastewater treatment facilities, or synthetic microbots in the cardiovascular system. In summary, the results presented here shed light on microbial transport and allow for the development of strategies for controlling surface rheotaxis.

## Methods

**3D tracking experiments**. The bacteria are smooth swimmers *E. coli* (CR20), a $\Delta$CheY mutant strain that almost never tumble. Suspension are prepared using the following protocol: bacteria are inoculated in 5 mL of culture medium (M9G: 11.3

g/L M9 salt, 4 g/L glucose, 1 g/L casamino acids, 0.1 mM CaCl$_2$, 2 mM MgSO$_4$) with antibiotics and grown overnight. In this way, bacteria with a fluorescently stained body are obtained. Then the bacteria are transferred in Motility Buffer (MB: 0.1 mM EDTA, 0.001 mM l-methionine, 10 mM sodium lactate, 6.2 mM K$_2$HPO$_4$, 3.9 mM KH$_2$PO$_4$) and supplemented with 0.08 g/mL L-serine and 0.005% polyvinyl pyrrolidone (PVP). The addition of L-serine increases the bacteria mobility and PVP is classically used to prevent bacteria from sticking to the surfaces. The interactions that come into play using this system are thus solely steric and hydrodynamic. After incubating for an hour in the medium to obtain a maximal activity, the solution was mixed with Percol (1:1) to avoid bacteria sedimentation. Under these conditions, the average swimming speed is $v_s = (26 \pm 4)\ \mu\text{m s}^{-1}$. For the experiments the suspension is diluted strongly such as to be able to observe single bacteria trajectories without interactions between bacteria.

The experimental cell is a rectangular channel made in PDMS using soft lithography techniques. The channel height is $H = 100\ \mu\text{m}$, the width $W = 600\ \mu\text{m}$ and its length is of several millimetres. Using a syringe pump (dosing unit: Low Pressure Syringe Pump neMESYS 290N and base: Module BASE 120N) we flow the suspension inside the channel at different flow rates (1, 1.9, 4.5, 9.0, 18, 50 nL s$^{-1}$), corresponding to wall shear rates of 1–50 s$^{-1}$. To have access to the 3D trajectories of single bacteria under flow we use a 3D Lagrangian tracker[47] which is based on real-time image processing, determining the displacement of a $xz$ mechanical stage to keep the chosen object at a fixed position in the observation frame. The $y$ displacement is based on the refocusing of the fluorescent object, keeping the moving object in focus with a precision of a few microns in $y$. The acquisition frequency is 30 Hz. The Lagrangian tracker is composed of an inverted microscope (Zeiss-Observer, Z1) with a high magnification objective (100×/0.9 DIC Zeiss EC Epiplan-Neofluar), a $xz$ mechanically controllable stage with a $y$ piezo-mover from Applied Scientific Instrumentation (ms-2000-flat-top-xyz) and a digital camera ANDOR iXon 897 EMCCD. Trajectories are only considered when far away from the lateral walls (distances larger than 100 μm) and as long as they are within 5 μm from the surface. A typical error on this distance, resulting from the uncertainty on the $y$-detection as well as the uncertainty on the position of the bottom surface is around 3 μm.

**Fluorescence experiments.** For flagella visualization, we use a genetically modified strain from the AB1157 wild-type (AD1) published in ref. [48]. This strain contains a FliC mutation to bind to the dye from Alexa Fluor. Single colonies of frozen stocks are incubated overnight (≈16 h) in 5 mL of liquid Luria Broth at 30 °C, shaken for aeration at 200 rpm. The bacteria are washed and resuspended in Berg's motility buffer (BMB: 6.2 mM K$_2$HPO$_4$, 3.8 mM KH$_2$PO$_4$, 67 mM NaCl and 0.1 mM EDTA). For flagella staining, 0.5 mL of the bacterial suspension in BMB at $2 \times 10^9$ bact/mL are mixed with 5 μL of Alexa Fluor 546 C5-maleimide suspended at 5 mg/mL in DMSO. The sample is kept in the dark, shaking at 100 rpm for 1 h. Bacteria are then three times washed in BMB and finally suspended at $10^8$ bact/mL in BMB supplemented with polyvinylpyrrolidone (PVP 350 kDa: 0.005%) to prevent sticking to the walls of the microchannel. The solution is then seeded with passive particles to be used for flow velocity determination (latex beads from Beckman Coulter $d = 2$ μm, density $\rho = 1.027$ g/mL at a volume fraction $\phi = 10^{-7}$).

The microfluidic PDMS channel is $H = 20$ μm deep, $W = 200$ μm wide and several millimetres long. We generate a flow by applying a precise pressure gradient, $\nabla p$, and we verify the shear rate by measuring the maximum flow rate $V_{\max}$ in the centre of the channel with tracer particles to guarantee an accurate measurement. We capture the bacterial dynamics far from the lateral walls and within 5 μm from the surface using an inverted microscope (Zeiss-Observer, Z1) with a high magnification objective (100×/0.9 DIC Zeiss EC Epiplan-Neofluar) and a digital camera ANDOR iXon 897 EMCCD at 30 fps. As bacteria are transported downstream, they are kept in the frame of observation by manually displacing the microscope's stage, which position is registered. During post-processing we extract the bacterial positions and orientations from the images.

**Data analysis.** To determine the frequency of the bacterial oscillations, we Fourier transform the bacterial trajectories obtained from experiments for different shear rates. (1) In the case of the experiments using bacteria with fluorescently labelled flagella, the in-plane angle $\psi(t)$ of the cell orientation is determined by fitting an ellipse to the acquired camera image and distinguish between head and tail by the velocity director. The uncertainty in this measurement depends on the aspect ratio of the swimmer and the pixel resolution of the experiment. For the trajectory shown in Fig. 1d, the length of the bacterium is $a = 26 \pm 2$ pixels, and its width is $b = 7 \pm 2$ pixels. Hence, the uncertainty of the in-plane angle is $\Delta\psi = \max_\pm \left(\arctan\left(\frac{a \pm \delta a}{b \pm \delta b}\right) - \arctan\left(\frac{a}{b}\right)\right) \approx 5.5$ degrees. This is indeed small compared with the observed oscillation amplitudes. Also note that the oscillation frequencies (1–10 Hz) are typically comparable or larger than the tumbling frequency (distributed about 1 Hz), and we only analysed trajectories in which no tumbles occurred, so sufficiently many oscillation periods are observed for a clean signal. (2) In the case of the 3D tracking experiments, we use the lateral velocity $v_z(t)$ to search for oscillatory motion. The uncertainty in this measurement also depends on the pixel resolution; $\Delta v_z = v_z \frac{\delta a}{a} \approx 1.5\ \mu\text{m s}^{-1}$. This is again small compared with the observed oscillation amplitudes. These uncertainties are shown as error bars in Fig. 1c, d, in the top right corners.

Hence, either $\psi(t)$ or $v_z(t)$ are Fourier transformed for trajectories of sufficiently long duration to resolve the lowest and highest frequencies accurately. The frequency of each trajectory is determined by selecting the highest peak in the resulting Fourier spectrum. This is repeated for all trajectories to form an ensemble of frequencies, from which we evaluate the mean frequency $f(\dot\gamma)$ and its standard deviation. For the fluorescence experiments, the data points at $\dot\gamma = 3.2, 12, 20, 32\ \text{s}^{-1}$ are averages over an ensemble of $N = 7, 7, 13, 7$ trajectories, each about 5 s in length to ensure a good signal from the oscillations in the Fourier transform. For the 3D tracking experiments, the data points at $\dot\gamma = 1.1, 1.9, 4.5, 9.0, 18, 49\ \text{s}^{-1}$ are averages over an ensemble of $N = 39, 18, 16, 9, 9, 3$ trajectories. These trajectories are much longer in time, around 30 s with a longest of 66 s.

The trajectories from the fluorescently labelled flagella were also used to compute the experimental distributions of in-plane orientation, PDF($\psi$). In Fig. 5c, the peak positions of these distributions are shown as magenta stars, where horizontal error bars stem from the uncertainty of the bacterial $z$ position (see the section 'Definition of shear rate' under Methods), and the vertical error bars correspond to $\Delta\psi^* \approx 20°$, the peak width.

**Definition of shear rate.** The flow profile in a rectangular channel of width $W$ and height $H$, at position $-\frac{W}{2} \leq z \leq \frac{W}{2}$ and $0 \leq y \leq H$, is given by

$$V(y,z) = \sum_{n\ \text{odd}}^{\infty} \frac{4H^2 \nabla p}{\pi^3 n^3 \mu} \left(1 - \frac{\cosh\left(\frac{n\pi z}{H}\right)}{\cosh\left(\frac{n\pi W}{2H}\right)}\right) \sin\left(\frac{n\pi y}{H}\right). \quad (6)$$

We perform all our experiments far from the side walls, $|z| \ll \frac{W}{2}$, such that this expression reduces to the planar Poiseuille flow $V(y) = \frac{H^2 \nabla p}{2\mu}\left(\frac{y}{H} - \frac{y^2}{H^2}\right)$, where $V_{\max} = \frac{H^2 \nabla p}{8\mu}$ is the maximum flow at the centre of the channel. Hence, the shear rate at the bottom surface is defined as

$$\dot\gamma = \frac{\partial V}{\partial y}\bigg|_{y=0} = \frac{4 V_{\max}}{H}. \quad (7)$$

Note that in our experiments the bacteria are within 5 μm from the surface. In the $H = 100$ μm channels this gives a 10% uncertainty in the shear rate, and in the $H = 20$ μm channels a 50% uncertainty. We have included these uncertainties as horizontal error bars in Figs. 1 and 5.

**Chirality-induced rheotaxis.** Marcos et al. used resistive force theory (RFT) to calculate the rheotactic behaviour of helical bacteria in shear flows in the bulk[17,46]. Based on their work and using their Mathematica notebook, that includes the RFT calculations for a passive helix subjected to shear flow, we identify the full angular dependence of the rheotactic torque of a swimming bacterium Eq. (2).

In RFT, a helical flagellum segment is approximated by a stiff slender rod with anisotropic friction coefficients $\xi_\perp$ and $\xi_\parallel$ with $1 < \xi_\perp/\xi_\parallel < 2$. The viscous force per unit length opposing the motion of a rod is written as $\boldsymbol{f} = -\xi_\parallel \boldsymbol{u}_\parallel - \xi_\perp \boldsymbol{u}_\perp$, with the local rod velocity $\boldsymbol{v}_l$ relative to the external shear flow ($\boldsymbol{v}_f = \dot\gamma y \hat{\boldsymbol{x}}$), $\boldsymbol{u} = \boldsymbol{v}_l - \boldsymbol{v}_f = \boldsymbol{u}_\parallel + \boldsymbol{u}_\perp$, where $\boldsymbol{v}_l$ is a sum of its translational and rotational velocity $\boldsymbol{v}_l = \boldsymbol{v} + \boldsymbol{\Omega} \times \boldsymbol{r}$ ($s;\psi,\theta$), and $\boldsymbol{u}_\parallel = (\boldsymbol{u}\cdot\hat{\boldsymbol{t}})\hat{\boldsymbol{t}}$, $\boldsymbol{u}_\perp = \boldsymbol{u} - \boldsymbol{u}_\parallel$; here $\boldsymbol{r}(s;\psi,\theta)$ is a space-curve of a helix parametrised by $s$, centred around $\boldsymbol{r} = 0$ and oriented along the swimmer direction $\boldsymbol{e} = (-\cos\theta\cos\psi, \sin\theta, -\cos\theta\sin\psi)$, and the tangent is $\boldsymbol{t} = (d\boldsymbol{r}/ds)/|d\boldsymbol{r}/ds|$. After integrating the total force and torque along the helix and averaging over the helix phase angle, one can solve for the unknown helix velocity $\boldsymbol{v}$ and angular velocity $\boldsymbol{\Omega}$. While $\boldsymbol{\Omega}$ is a good approximation of the Jeffery reorientation rate, the velocity $\boldsymbol{v}$ determines a chirality-induced migration velocity, which only vanishes for non-chiral particles. For swimming bacteria the head acts as an anchor that induces, in addition to $\boldsymbol{\Omega}$, a rheotactic torque $\boldsymbol{\Omega}^H = -k_2 \boldsymbol{e} \times \boldsymbol{v}$, where the prefactor $k_2$ depends on the shape of the cell body[46].

The analytic expressions for $\boldsymbol{v}$ obtained with Mathematica are very long, but we attach the full expressions in the supplementary file "ChiralityRheotaxisExpressions.dat" which can be imported again with the `Get [filename]` command. As shown in the supplementary Mathematica notebook file "ComparisonGraphicalFormula.nb", we import and plot the velocity components $v_x$, $v_y$ and $v_z$ depending on the orientation angles $\psi$ and $\theta$ for a given helix shape. Hence, we demonstrate that the exact angular dependences can be simplified to

$$v_x = -k_1 \dot\gamma \sin 2\psi \cos\theta^2, \quad (8a)$$

$$v_y = -k_1 \dot\gamma \sin\psi \sin 2\theta, \quad (8b)$$

$$v_z = 2k_1 \dot\gamma(-\sin^2\psi \cos^2\theta + \cos 2\theta) \quad (8c)$$

which linearly increase with the shear rate $\dot\gamma$, and where the prefactor $k_1$ only depends on the helix geometry. Importantly, $k_1 > 0$ for a left-handed helix, as it is the case for the normal form of E. coli bacteria, and <0 otherwise. The components $\Omega_x^H$, $\Omega_y^H$ and $\Omega_z^H$ of the rheotactic angular velocities are then given by

$$\Omega_x^H = -\bar\nu_H \dot\gamma(\cos 2\theta - 2\cos^2\theta \sin^2\psi), \quad (9a)$$

$$\Omega_y^H = -\bar{\nu}_H \dot{\gamma}(\cos\theta + \cos 3\theta)\cos\psi/2, \qquad (9b)$$

$$\Omega_z^H = -\bar{\nu}_H \dot{\gamma}\cos^2\theta \sin\theta \sin 2\psi, \qquad (9c)$$

where $\bar{\nu}_H = 2k_1 k_2$. The components of this torque in the $\psi$ and $\theta$ directions are written down in Eq. (2).

**BD simulations of surface rheotaxis.** The bacterial surface rheotaxis is simulated by numerical integration of the orientation dynamics, encapsulated by the covariant Langevin equation[67] written out in terms of the angles $(\psi, \theta)$ that live on the curved surface $|\mathbf{p}| = 1$,

$$\delta\psi = (\Omega_\psi^W + \Omega_\psi^F + \Omega_\psi^V)\delta t + \frac{\sqrt{2D_r \delta t}}{\cos\theta}\eta_\psi, \qquad (10a)$$

$$\delta\theta = (\Omega_\theta^W + \Omega_\theta^F + \Omega_\theta^V)\delta t - \tan\theta D_r \delta t + \sqrt{2D_r \delta t}\eta_\theta, \qquad (10b)$$

where $D_r = 0.057\ \mu m^2/s$ is the rotational diffusion coefficient[61], the noise correlations are defined as $\langle\eta_i\rangle = 0$ and $\langle\eta_i(t)\eta_j(t')\rangle = \delta_{ij}\delta(t - t')$ and the deterministic terms are written out explicitly in Eqs. (11a) and (11b) below.

At the start of a surface trajectory, the swimmer is initiated parallel to the surface, $\theta(t = 0) = 0$, and with a random uniformly distributed in-plane angle, $\psi(0) \in [-\pi, \pi]$. Subsequently, its orientation $(\psi(t), \theta(t))$ is integrated using a forward Euler scheme with time step $\delta t = 10^{-3}$ s, and after every time step the orientation angles are renormalised to their domains, $\psi \in [-\pi, \pi]$ and $\theta \in [-\pi/2, \pi/2]$, by setting $\psi \to \mathrm{mod}(\psi, 2\pi, -\pi)$ and $\theta \to \mathrm{abs}(\theta + \pi/2) - \pi/2$. Next, the spatial dynamics is obtained by computing the velocity parallel to the wall, $\mathbf{v}_\parallel = v_0 \mathbf{p}_\parallel + \dot{\gamma} h_s \hat{\mathbf{x}}$, where $v_0 = 20\ \mu m/s$ is the swimming speed, $\mathbf{p}_\parallel = p_x \hat{\mathbf{x}} + p_z \hat{\mathbf{z}} = (-\cos\theta\cos\psi, -\cos\theta\sin\psi)$ is the swimmer orientation parallel to the wall, and $h_s = 1\ \mu m$ is the distance from the wall. The surface trajectories are then found by numerical integration of $\dot{\mathbf{r}}_\parallel = \mathbf{v}_\parallel$. A trajectory ends when the pitch angle exceeds the escape angle, $\theta_E = 20°$, after which the swimmer escapes the surface.

Hence, $N = 10^4$ trajectories are simulated for each value of the applied shear rate. For the orientation distributions we used $\dot{\gamma} = 0$ and $\dot{\gamma} = 10^{2.5(i-1)/9}$, where $i = 1, 2, \ldots, 9, 10$. From these trajectories $(\psi, \theta)$ the distributions PDF$(\psi)$ and PDF$(\theta)$ follow immediately. For the frequency analysis we also looked at lower shear rates, $\dot{\gamma} = 10^{-1+3.5(i-1)/19}$, where $i = 1, 2, \ldots, 19, 20$. To determine the frequency of the bacterial oscillations, a trajectory must be sufficiently long to resolve the smaller frequencies. At high shear the average trajectory (residence) time is smaller than at low shear, so we discard trajectories shorter than 10 s, which leaves at least ~100 trajectories for any shear rate. The frequencies of the remaining trajectories are then obtained individually by Fourier transforming the in-plane angle $\psi(t)$, and selecting the frequency of the highest peak in the resulting Fourier spectrum. Using this ensemble, we evaluate the mean frequency $f(\dot{\gamma})$ and its standard deviation. Hence, the error bars in Fig. 1f correspond to two standard deviations from the ensemble mean, the 95% confidence interval.

**Estimation of basis parameter set.** Hydrodynamic and steric surface realignment, $\nu_W$: Both steric and hydrodynamic interactions can contribute to reorient swimmers with respect to walls. We model this surface alignment with a generic functional form, $\Omega_\theta^W(\theta) = -\nu_W \sin 2(\theta - \theta_0)$, noting that the double angle sine has been used before in the literature as descriptions for both steric and hydrodynamic interactions[21,51]. Accordingly, the prefactor is estimated considering both contributions: The reorientation rate $\nu_W$ due to steric interactions has been reported to be of the order $\nu_W^{steric} \sim 1 - 10\ s^{-1}$ for flagellated *Caulobacter* bacteria[21]. Reorientation rates due to hydrodynamic interactions, away from the walls, can be approximated by far-field expressions[51] with a prefactor $\nu_W^{hi} = 3p/(128\pi\eta h^3)$, where $p = 0.8$ pN $\mu$m is the dipole strength of *E. coli* bacteria[61], $\eta = 10^{-3}$ Pa s the viscosity of water, and $h$ the distance of the swimmer from the surface. Using these expressions with $h \approx 1-2\ \mu m$, one finds $\nu_W^{hi} \approx 0.75 - 6\ s^{-1}$. This rate is smaller than the steric contribution, but also acts when the swimmer moves away from the surface, so it could increase the wall residence time[62]. Also note that the $1/h^3$ dependence likely gives an overestimate when a swimmer is very close to the wall, $h < 1\ \mu m$, as the multipole approximation breaks down. Taking this information together, we use $\nu_W = 4\ s^{-1}$ to capture the combined effects of steric and hydrodynamic interactions.

Zero-shear pitch angle, $\theta_0$: Recent results have shown that the zero-shear pitch angle is not exactly parallel to the wall ($\theta = 0$) but sightly points into the surface[49]. We have included this in our model through the surface alignment term, with the functional form $\sin(2(\theta - \theta_0))$, with $\theta_0 = -10°$, the measured mean value.

Swimmer aspect ratio $\Gamma$: While cell bodies of *E. coli* bacteria have typical aspect ratios of $\Gamma \approx 3$, the flagellar bundle increases the effective aspect ratio, so we choose $\Gamma = 4$. Note that $\Gamma$ does not enter the model directly, but rather $G = \frac{\Gamma^2 - 1}{\Gamma^2 + 1} \lesssim 1$, which does not change significantly with $\Gamma$.

Circling near a wall $\nu_C$: The typical circling frequencies of an *E. coli* bacterium close to a boundary is on the order of ~1 s, which gives typical circles of radius $R = v_0/\Omega_\psi^W \sim 20\ \mu m$[31,52]. Therefore, we choose $\nu_C = 1\ s^{-1}$. Of course, there is quite a variety in these frequencies between different individual bacteria[52,53], which

will lead to bacteria having different individual critical shear rates. Also note that most bacteria have left-handed flagella, which gives rise to clock-wise circles $v_C > 0$ on surfaces as seen from the liquid side, but for bacteria with right-handed flagella the prefactor would be negative, $v_C < 0$.

Chirality-induced reorientation rate $\bar{\nu}_H$: The rheotactic drift for bacteria in bulk has been quantified by Marcos et al.[17,46], as described in the section 'Fluorescence experiments'. We expect a similar rheotactic strength for our bacteria, approximately $\bar{\nu}_H = 0.02$.

Weathervane reorientation rate $\bar{\nu}_V$: The weathervane effect stems from the idea that the cell head experiences hydrodynamic friction with the surface. This can be seen as an effective anchoring of the cell body compared with the flagella, such that the tail is dragged along with the flow, like a flag in the wind. The prefactor of the weathervane effect lies between zero and unity, $\bar{\nu}_V \in [0, 1]$, which corresponds to the strength of the hydrodynamic friction, and also the relative size of the tail compared with the body. A detailed analysis based on lubrication theory was performed to explain this for sperm cells[34], and experimentally they find $\bar{\nu}_V^{sperm} \approx 0.12$. Because of their similar cell body size but much shorter tail, we expect that the prefactor for bacteria should be larger compared with sperm, so we estimate $\bar{\nu}_V^{bact} \approx 0.5$.

Weathervane angle $\theta_V$: The magnitude of the weathervane effect is reduced strongly when the swimmer is not in close proximity to the walls, since it relies on enhanced friction obtained from the lubrication regime[34,59]. Therefore, we expect the anchoring strength to vanish quickly when the swimmer is oriented away from the surface, when $\theta > 0$. To account for this, we introduce a tanh-function Eq. (3) with a small value $\theta_V = 0.04 = 2.3°$, so that the weathervane effect vanishes when the cell is pointing away from the surface.

Downstream advection: To calculate the downstream advection velocity due to the linear shear flow, we use the distance from the surface $h_s = 1\ \mu m$, so the velocity is $\mathbf{v}_f = \dot{\gamma} h_s \hat{\mathbf{x}}$. If the cells are oriented upstream they can also move against weak flows. This is seen in Fig. 3, where we use a constant swimming speed $v_s = 20\ \mu m\ s^{-1}$. Note that these parameters do not affect the orientation dynamics in our model.

Escape angle $\theta_E$: In order to determine when a swimmer leaves a surface, it has to reach a certain escape angle $\theta_E$. This model allows for the possibility of hydrodynamic surface attraction[20] by choosing a slightly positive escape angle, such that bacteria remain hydrodynamically trapped even when they are slightly oriented away from the surface. We choose to be $\theta_E = 20°$, following refs. [61–63]. Note that $\theta_E$ also does not influence the dynamics in the model per se, but rather defines how long a bacterium stays at a surface.

Rotational diffusion: The orientation of the bacteria is affected by fluctuations as they swim. We use the rotational diffusion coefficient $D_r = 0.057\ \mu m^2\ s^{-1}$[61].

Tumbling: We do not include tumbling in our BD simulations presented in the main text because we use smooth swimmers in our 3D tracking experiments, but in Supplementary Fig. 2 we show orientation distributions for simulations that include tumbles. We simulate these tumbling events by temporarily increasing the swimmer's rotational diffusion coefficient[63], to $D_T = \varphi^2/(2\tau_T)$, where the average tumble angle $\varphi = \pi/3 = 60°$[68], the tumbling time is $\tau_T = 0.1$ s, and the duration between tumbles is exponentially distributed with average run time $\tau_R = 1$ s.

**Robustness in parameter variations.** Whereas the basis set of our model parameters has been estimated carefully from experiments and the literature (see section above), we demonstrate here that our results are robust with respect to variations in these quantities. In Table 1 we list the critical shear rates for a wide range of parameters, as well as the optimal shear rate for upstream orientation, $\dot{\gamma}_u$, and the corresponding in-plane angle, $\psi_u = \psi_R^*(\dot{\gamma}_u)$, evaluated at the right fixed point. Note that all these values were found numerically from the deterministic model Eqs. (11a) and (11b), without linearising or using other approximations. In Supplementary Fig. 3, we also show the full orientation dynamics and the associated fixed point behaviour for a wide range of shear rates.

Importantly, all four rheotaxis regimes (I–IV) exist for all parameter values tested, and their critical shear rates only shift up or down. Swimming to the right is always stable, and swimming to the left is always stable above the third critical shear rate, for all parameter values tested. In some cases at high shear rates the fixed point becomes locally unstable, but the same dynamics is still observed because a stable limit cycle forms around the fixed point. In the following we discuss the effect of each parameter specifically.

Surface realignment, $\nu_W$: Most bacterial strains are known to align with surfaces, including *E. coli* and *Caulobacter* but also sperm cells and other microbes. The strength of surface alignment, $\nu_W$, does not affect the first transition $\dot{\gamma}_{c_1}$. However, the other critical shear rates increase with rising $\nu_W$, and it also improves (decreases) the optimal upstream orientation $\psi_u$. Physically this makes sense, as a strong alignment to $\theta_0$ suppresses oscillations and provides better anchoring for the weathervane effect.

Pitch angle, $\theta_0$: A deeper pitch angle provides better anchoring for the weathervane effect, so the optimal upstream angle $\psi_u$ improves and oscillations can only start at larger shear rates. As expected, the residence time at the surface also increases with a deeper pitch angle and a stronger $\nu_W$ prefactor.

Circling strength, $\nu_C$: The circle-swimming torque affects the Adler transition strongly. When $\nu_C$ is small compared with the weathervane effect, the first threshold $\dot{\gamma}_{c_1}$ is small as swimmers break out of their circles at low shear. The effect

## Table 1 Robustness of the model with respect to parameter variations

| | $\dot{\gamma}_{c_1}$ | $\dot{\gamma}_{c_2}$ | $\dot{\gamma}_{c_3}$ | $\dot{\gamma}_u$ | $\psi_u$ |
|---|---|---|---|---|---|
| Basis | 1.53 | 24.4 | 44.2 | 16.4 | 10.8° |
| $\nu_W = 0.5\,\mathrm{s}^{-1}$ | 1.70 | 2.4 | 13.5 | 3.6 | 51° |
| $\nu_W = 10\,\mathrm{s}^{-1}$ | 1.52 | 65 | 88 | 36 | 5.9° |
| $\theta_0 = -20°$ | 1.23 | 43 | 81 | 29 | 6.4° |
| $\theta_0 = 0°$ | 4.2 | 6.8 | 19 | 9.4 | 71° |
| $\nu_C = 0.2\,\mathrm{s}^{-1}$ | 0.31 | 26 | 34 | 13 | 4.1° |
| $\nu_C = 10\,\mathrm{s}^{-1}$ | 17.5 | 21 | 127 | 34 | 58° |
| $\Gamma = 1$ | 1.73 | 3.5 | 9.1 | 3.1 | 48° |
| $\Gamma = 8$ | 1.52 | 89 | 348 | 49 | 5.8° |
| $\bar{\nu}_H = 0.002$ | 1.55 | 25 | 36 | 18 | 8.4° |
| $\bar{\nu}_H = 0.1$ | 1.52 | 16 | 235 | 13 | 29° |
| $\bar{\nu}_V = 0.1$ | 3.9 | 23 | 38 | 17 | 32° |
| $\bar{\nu}_V = 1.0$ | 0.88 | 25 | 52 | 17 | 6.1° |
| $\theta_V = 0.57°$ | 1.54 | 23 | 38 | 19 | 8.4° |
| $\theta_V = 5.7°$ | 1.59 | 27 | 56 | 17 | 13° |

Columns give the three critical shear rates, $(\dot{\gamma}_{c_1}, \dot{\gamma}_{c_2}, \dot{\gamma}_{c_3})$ $(\mathrm{s}^{-1})$, the optimal shear rate for upstream orientation, $\dot{\gamma}_u$ $(\mathrm{s}^{-1})$, and the corresponding in-plane angle at the right fixed point, $\psi_u = \psi_R^*(\dot{\gamma}_u)$. The first row corresponds to the basis parameter set, and subsequent rows vary one parameter at a time, as indicated in the first column, keeping the other parameters from the basis set: $\nu_W = 4\,\mathrm{s}^{-1}$, $\theta_0 = -10°$, $\nu_C = 1\,\mathrm{s}^{-1}$, $\Gamma = 4$, $\bar{\nu}_H = 0.02$, $\bar{\nu}_V = 0.5$, $\theta_V = 2.3°$

on the $\dot{\gamma}_{c_2}$ transition is not really significant because this term only contributes to the $\Omega_\psi$ and not the $\Omega_\theta$ dynamics. However, a strong torque $\Omega_\psi$ can upset the stability of the left fixed point at lower shear rates. Moreover, it can shift the right fixed point to larger $\psi^*$ values, especially at low shear where the terms proportional to $\dot{\gamma}$ are weak, such that the upstream swimming capacity is reduced.

Aspect ratio, $\Gamma$: Different bacterial species feature a wide range of aspect ratios, from almost spherical cells with $\Gamma = 1$ to rodlike cyanobacteria with $\Gamma > 100$. The aspect ratio affects the geometric factor, $G = \frac{\Gamma^2 - 1}{\Gamma^2 + 1} \lesssim 1$. This changes the strength of the Jeffery orbit because of its approximate $\Omega_\theta^J \propto (1 - G)$ dependence for small $\theta$ values. Therefore, small aspect ratios give stronger orbits, so that the oscillations emerge for smaller shear rates compared with the basis set. The Adler transition $\dot{\gamma}_{c_1}$ is not affected much by this term. Note though that other parameters like $\bar{\nu}_H$ and $\bar{\nu}_V$ also depend on the cell geometry and aspect ratio.

Helix chirality-induced rotation, $\bar{\nu}_H$: The effect due to the chirality of the bacterial helix is rather humble, because its value is expected to be small $\bar{\nu}_H \ll 1$. Even for very large values the transitions $\dot{\gamma}_{c_1}$ and $\dot{\gamma}_{c_2}$ do not change much. Still, it does introduce a bias towards swimming to the right, which in bulk leads to a significant symmetry breaking. Also on the surface we see that a large $\bar{\nu}_H$ increases the third critical shear rate, as expected.

Weathervane effect, $\bar{\nu}_V$: The weathervane prefactor signifies the strength of surface anchoring, the hydrodynamic friction that the cell head derives from the surface compared with the cell tail. Lubrication theory[34] has shown that this term increases with cell body size, but decreases with flagellar length. This strongly affects the Adler transition $\dot{\gamma}_{c_1}$, because the upstream bias breaks swimmers out of their circular trajectories. Indeed, similar to the circling term, this does not affect $\dot{\gamma}_{c_2}$ nor $\dot{\gamma}_{c_3}$ much, but a stronger anchoring improves the optimal angle for upstream motion $\psi_u$.

Weathervane sensitivity, $\theta_V$: The angle over which the anchoring term switches from strong ($\theta < \theta_V$) to weak ($\theta > \theta_V$) is called the sensitivity of the weathervane effect. This parameter does not change the critical shear rates significantly, nor the upstream swimming potential. Still, it modifies the oscillation frequency a bit see Eq. (23) because the on-off switching of the weathervane effect can help drive the oscillations (Fig. 4).

**Summary of equations of motion**. Combining the deterministic contributions of our rheotactic model, we have the reorientation rates

$$\Omega_\psi = \nu_C + \frac{\dot{\gamma}}{2}(1 + G)\sin\psi\tan\theta$$
$$+ \dot{\gamma}\bar{\nu}_H\cos\psi\frac{\cos 2\theta}{\cos\theta} - \dot{\gamma}\bar{\nu}_V\sin\psi\frac{1 - \tanh\frac{\theta}{\theta_V}}{2}, \quad (11a)$$

$$\Omega_\theta = -\nu_W\sin 2(\theta - \theta_0) + \frac{\dot{\gamma}}{2}(1 - G\cos 2\theta)\cos\psi$$
$$+ \dot{\gamma}\bar{\nu}_H\sin\psi\sin\theta - \dot{\gamma}\bar{\nu}_V\sin\theta\frac{1 - \tanh\frac{\theta}{\theta_V}}{2}. \quad (11b)$$

The $\bar{\nu}$ are dimensionless and the $\nu$ without bar have units $(\mathrm{s}^{-1})$.

**Adler transition**. The first transition, from circling to straight motion, has previously been described in the literature for sperm cells[34]. It can be characterised as

the point at which the upstream-directed torque from weathervane effect becomes more important than the constant torque from the bacterial circling on surfaces. This can be captured in the limit of small pitch angles, $\theta \to 0$, where the one-dimensional equation for the in-plane angle Eq. (11a) simplifies to

$$\Omega_\psi = \nu_C - \frac{1}{2}\dot{\gamma}\bar{\nu}_V\sin\psi + \dot{\gamma}\bar{\nu}_H\cos\psi. \quad (12)$$

This can immediately be solved for the equilibrium angle $\psi^*$ where $\Omega_\psi = 0$. The solution is a little long to write down, but is nothing more than an arctangent. Further progress can be made by noting that $\bar{\nu}_H \ll \bar{\nu}$ and also $\cos\psi \ll \sin\psi$, near $\psi \sim \pi/2$, at shear rates close to the transition where the bacteria swim to the right. Consequently, the resulting equilibrium angle is given by

$$\psi^* = \arcsin\frac{2\nu_C}{\bar{\nu}_V\dot{\gamma}}. \quad (13)$$

This function has no real solutions for shear rates smaller than a critical value. Indeed, the rotating bacteria do not have a stable equilibrium orientation, but for large enough shear they break out of their circles and maintain a constant bearing. Therefore, we find the critical shear rate of Adler transition,

$$\dot{\gamma}_{c_1}^{th} = \frac{2\nu_C}{\bar{\nu}_V}. \quad (14)$$

For finite pitch angles, $\theta \to \theta_0$, this derivation is extended straightforwardly and gives the first critical shear rate,

$$\dot{\gamma}_{c_1}^{th} = \frac{2\nu_C}{\bar{\nu}_V - (1 + G)\tan(\theta_0) - \bar{\nu}_V\tanh\left(\frac{\theta_0}{\theta_V}\right)}. \quad (15)$$

For the parameters used this gives $\dot{\gamma}_{c_1}^{th} \sim 1.50$, compared with the value from BD simulations $\dot{\gamma}_{c_1}^{sim} \sim 1.5$. The corresponding equilibrium angle is given by $\psi^* = \arcsin(\dot{\gamma}_{c_1}^{th}/\dot{\gamma})$, which is shown as the dotted green line in Fig. 5c.

**Approximation of the equilibrium orientations**. In order to estimate the equilibrium rheotactic orientations at higher shear rates above the Adler transition, the one-dimensional approach breaks down because the pitch angle $\theta$ becomes significant. This shortcoming is also observed from the 1D solution Eq. (13) that only decreases with increasing shear but does not capture the later increase, which gives rise to an optimum shear rate for upstream swimming.

Therefore, we aim to solve for both the angles $\psi^*$ and $\theta^*$, the fixed points such that $\Omega_\psi = \Omega_\theta = 0$. Because it is known that the bacteria swim to the right and left at high shear rates, we employ the ansatz

$$(\psi^*, \theta^*) = \left(\varepsilon_\psi \pm \frac{\pi}{2}, \varepsilon_\theta\right), \quad (16)$$

where $\varepsilon_\psi^R, \varepsilon_\theta^R$ and $\varepsilon_\psi^L, \varepsilon_\theta^L$ are small and where the top sign (+) corresponds to swimming to the right and the bottom sign (−) corresponds to swimming to the left. Inserting this expression into the equations of motion Eqs. (11a) and (11b) and expanding to first order in $\varepsilon_\psi$ and $\varepsilon_\theta \ll 1$ yields the linear expression

$$\begin{bmatrix} 0 \\ 0 \end{bmatrix} = \begin{bmatrix} \Omega_\psi \\ \Omega_\theta \end{bmatrix} = \begin{bmatrix} E \\ F \end{bmatrix} + \begin{bmatrix} A & B \\ C & D \end{bmatrix}\begin{bmatrix} \varepsilon_\psi \\ \varepsilon_\theta \end{bmatrix} + \dots, \quad (17)$$

where the shear-dependent coefficients are

$$A = \mp\bar{\nu}_H\dot{\gamma}, \quad (18a)$$

$$B = \pm\frac{1}{2}\dot{\gamma}(1 + G) \pm \frac{\bar{\nu}_V\dot{\gamma}}{2\theta_V}, \quad (18b)$$

$$C = \mp\frac{1}{2}\dot{\gamma}(1 - G), \quad (18c)$$

$$D = -2\nu_W\cos(2\theta_0) - \frac{1}{2}\bar{\nu}_V\dot{\gamma} \pm \bar{\nu}_H\dot{\gamma}, \quad (18d)$$

$$E = \nu_C \mp \frac{1}{2}\bar{\nu}_V\dot{\gamma}, \quad (18e)$$

$$F = \nu_W\sin(2\theta_0). \quad (18f)$$

This matrix equation can be inverted directly, so Eq. (17) can be solved to find the perturbations $\varepsilon_\psi$ and $\varepsilon_\theta$, and hence the equilibrium in-plane and pitch angles,

$$\psi^* = \frac{DE - BF}{BC - AD} \pm \frac{\pi}{2}, \quad (19a)$$

$$\theta^* = \frac{CE - AF}{AD - BC}. \quad (19b)$$

These results are shown as the dashed lines in Fig. 5c, d, where they may be compared with the positions of found in the experiments (magenta stars), the Brownian dynamics simulations (points), and the positions of the true fixed points (solid lines) obtained numerically from the full deterministic model Eqs. (11a) and (11b). We use Mathematica's built-in function FindRoot[] to obtain these fixed

points numerically, with 1000 randomly distributed initial positions to ensure we capture all the points on the left and on the right. We have verified that this method is accurate with other solver algorithms and the `NSolve[]` function.

**Oscillation frequency**. We start with a linear stability analysis of the equilibrium orientation angles, $(\psi, \theta) = (\psi^* + \alpha, \theta^* + \beta)$, where $\alpha, \beta \ll 1$. Expanding the linearised equations of motion Eq. (17) to first order in $\alpha$ and $\beta$ then yields

$$\begin{bmatrix} \Omega_\psi \\ \Omega_\theta \end{bmatrix} \approx \begin{bmatrix} A & B \\ C & D \end{bmatrix} \begin{bmatrix} \alpha \\ \beta \end{bmatrix}, \qquad (20)$$

in terms of $(A, B, C, D)$ given by Eqs. (18a) through (18f). The eigenvalues of this matrix are

$$\lambda = \frac{1}{2} \left( A + D \pm \sqrt{(A-D)^2 + 4BC} \right). \qquad (21)$$

The real part of the eigenvalues characterises the stability of the equilibrium orientation. When it is negative the equilibrium is stable, mainly because of the surface alignment ($\nu_W$ term). This occurs for swimming to the right in regimes II onwards, and also for motion to the left in the fourth regime.

The imaginary part, however, characterises the presence of oscillations. For small shear rates the imaginary part is zero, but above a certain shear rate oscillations emerge. This occurs when $(A - D)^2 = 4BC$, at the critical shear rate

$$\dot{\gamma}_{c_2}^{\text{lin}} = \frac{4\nu_W \cos(2\theta_0)}{2\sqrt{(1 - G^2) + (1 - G)\bar{\nu}_V/\theta_V - \bar{\nu}_V \pm 4\bar{\nu}_H}}. \qquad (22)$$

Below this shear rate value, the equilibrium orientation is a "star-type" stable fixed point, whereas above $\dot{\gamma}_{c_2}$ it is a "spiral-type" stable fixed point with damped oscillations. The imaginary part of the eigenvalues Eq. (21) then gives the oscillation frequency,

$$f = \frac{1}{2\pi} \text{Im}(\lambda), \qquad (23)$$

which is shown as the dashed yellow line in Fig. 1f. For high shear rates, this tends to the linear function for the oscillation frequency given in Eq. (5). If the weathervane and helix chirality coefficients are small, $\bar{\nu}_V, \bar{\nu}_H \ll 1$, this further simplifies to the pure Jeffery frequency $\omega_O = \dot{\gamma}\sqrt{(1 - G^2)}/2$.

## Data availability

The data that support the plots within this paper and other findings of this study are available from the corresponding author upon request. The computer code for the BD simulations is also available upon request.

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

## Acknowledgements

We are grateful to the authors of ref. [46], and in particular to Henry Fu, providing the Mathematica notebook, including the resistive force theory calculations for a helix subjected to shear flow. We thank Angela Dawson and Vincent Martinez for providing the AB1157 *E. coli* strain. A.M. acknowledges funding from the Human Frontier Science Program (Fellowship LT001670/2017). E.C., N.F.M., A.L. and G.J. acknowledge funding from the ANR-15-CE30-0013 BacFlow. A.L., A.Z. and N.F.M. acknowledge funding from the ERC Consolidator Grant PaDyFlow (Agreement 682367). A.Z. acknowledges funding from the European Union's Horizon 2020 research and innovation programme under the Marie Skłodowska-Curie grant (Agreement 653284), and funding from the Austrian Science Fund (FWF) through a Lise-Meitner Fellowship (Grant No. M 2458-N36). This work has received the support of Institut Pierre-Gilles de Gennes (Équipement d'Excellence, "Investissements d'avenir", program ANR-10-EQPX-34).

## Author contributions

A.M., N.F.M., G.J., E.C., A.L. and A.Z. designed the research, N.F.M. and G.J. performed the experiments, A.M., N.F.M. and G.J. analysed the experimental data, A.M. and A.Z. developed the theory and simulations, A.M., N.F.M., A.L. and A.Z. wrote the paper.

## Additional information

**Competing interests:** The authors declare no competing interests.

