## [Peer Review File · Nature Communications]

Reviewer #1 (Remarks to the Author):

Oscillatory surface rheotaxis of swimming E. coli bacteria

In this paper, the authors observe experimentally a range of rheotactic behaviors for E. coli bacteria swimming next to a wall at different shear rates, using a novel Lagrangian-tracking technique allowing them to acquire long trajectories. The experimental observations are explained theoretically by combining previously known reorientation mechanisms of a bacterium with a new chirality-induced effect. Their theoretical approach includes simulations of individual trajectories and analytical results. The authors identify a new component of rheotaxis, that is oscillation of the orientation vector at high frequency for high shear rate. They also observe both in experiments and theory trajectories biased towards the negative vorticity on top of the established positive vorticity rheotaxis. Overall, this paper combines previously explored elements to provide a complete picture of bacteria rheotaxis in varying shear rates.

While a convincing match between the experimental and theoretical description of rheotaxis for different shear rates has been achieved, the degree of novelty remains less clear and the claim that this can be a novel approach to prevent contamination is not well substantiated. It would also be important to get clarification on some of the theoretical arguments used.

Impact:

- The suggested device for preventing contamination does not seem to utilize the presence of newly observed leftward trajectories/oscillations. Rather, it is based on the known tendency of the swimmers to move right when swimming upstream (Hill et al. 2007 in the reference list). As written, it is not clear that a novel mechanism for contamination prevention has been proposed.
- To follow up on the previous point, the novel observation of oscillations in trajectories and rheotaxis toward negative vorticity happen in a regime of shear where cells are advected downstream, so when their risk of swimming upstream is absent anyway. Therefore, one could argue that the knowledge of this regime of swimming is not directly relevant for contamination prevention.
- Overall, to support a claim that this study could help design devices deterring upstream swimming in medical devices or other applications, the authors should provide a more convincing case. This should include, in particular, estimates of common shear rates in medical devices, and which regimes of rheotaxis are expected in these applications.
- Similarly, could the authors discuss how their results obtained with a specific mutant of E. coli could extend to other species of bacteria? This would broaden the impact of their work.

Modelling:

- The model has about 10 parameters. While the match with the experimental data is valuable, some of the parameter values were chosen quite arbitrarily. To avoid over-fitting, can the authors provide evidence that the model predictions are robust against variation in the parameters, for example (but not only) θ_0 ?
- Could the authors clarify the derivation of Eq. 2? It appears to be the only new term in the overall orientational dynamics and its importance is stressed throughout the paper. Yet it is presented as a trial and error guess from some more complicated expression, which is not even shown. Since novelty is claimed here, more details should be given and even though Mathematica cannot simplify the long equation, it should nevertheless be displayed. In particular, since the authors write that 'in a good approximation the helix will rotate in flow similar as a rigid rod-like particle', isn't Eq. 2 effectively similar to Jeffrey's equation?
- Fixed points. Section 5c on the equilibrium orientations is written in a manner that suggests that $(\psi, \theta) = (\pi/2, 0)$ is a fixed point or is close to some true fixed point of the equations of motion M8 and M9. The authors look for the equilibrium orientation by setting $\Omega_\psi = \Omega_\theta = 0$ and propose $(\pi/2, 0)$ as an approximation because 'it's known that bacteria swim to the right and left at high shear rates'. The resulting solution of the linearized equations is then called an equilibrium solution while M14 clearly shows that there is still a torque $[\tau = (E, 0)$ for $(\psi, \theta) = (\pi/2, 0)$] in the ψ direction. There is no attempt to demonstrate how close $(\psi, \theta) = (\pi/2, 0)$ and the 1st order correction are to the true fixed point for relevant parameter values. Are there even any other fixed points?
- It should be clarified that θ_0 in Eq. M16 is not the same as the equilibrium angle with no flow (Wall effects). Right now, the notation is confusing: shouldn't the authors use ψ^* and θ^* in M16? Furthermore, for self-consistency, the authors should display θ^* in plot 4(b) or in another SI figure to check if θ^* converges to 0 as the shear vanishes, which is expected from the $\theta_0 = 0$ assumption of the 'Wall effects' section. It would also be interesting to verify the match between linearisation approach and numerical solution for this parameter θ^* in analogy to Fig. 4(b) for ψ_0 .

Minor comments:

- Beginning of second paragraph of the paper: "surface locomotion" could be confused with mechanism such as gliding, which are different from the flagella-based swimming above a surface described here. Reformulate.
- Second paragraph from Experimental Observation section: the explanations are unclear, because it is difficult to identify the trajectories that are mentioned on the corresponding plot. We would recommend mentioning the color of the trajectories as it is done in the Discussion section.
- Could the authors provide some idea of how frequently they observed cells swimming to the left with respect to right-swimming cells in the experiments? Right now, there no information on the experimental statistics.
- In the experiment with fluorescently tagged flagella: the shear rate could vary by 50% between the wall and the maximum tracked position of 5 μm . Could the authors elaborate on how would such a variation of shear rate modify the dynamics?

- Do plots/pictures in Figs. 1 (c,d) correspond to any particular trajectory shown in Fig. 1 (b)?
- Panel e of Figure 1: could the authors provide the statistics behind each experimental point (how many trajectories for each point, from how many independent experiments?).
- Figure 3: Confusion in the legend, which talks of “upper panels” and “lower panels” when the figure is organised in “left” and “right” panels.
- The direction of axes in Fig. 3 should match that of Fig. 1 - for example, Fig. 1 suggests that swimming upstream corresponds to swimming in the negative x-direction; the opposite is true for Fig. 3.
- In the equation for Ω^W_{ψ} just before paragraph “b.Flow effects”: a closing parenthesis is missing (same in equation M8?).
- What is the method used to find the numerical solutions of the fixed points?
- In Section “Methods/3D tracking experiments”: the concentrations of L-serine and PVP used in the motility buffer are missing.
- In Section “Methods/3.Simulations of surface rheotaxis”, the distance of the bacterium to the surface is fixed at value “ $\delta = W/2 = 0.5\mu\text{m}$ ”, but in “METHODS/4.Estimations of the parameters”, this distance is set to “ $h_s = 1\mu\text{m}$ ”, and in Fig. 3 caption, “ $h_s = 0.5\mu\text{m}$ ”. Which notation and value is correct? Please make notation consistent. In addition, I would suggest to replace “ $v_f = \gamma y \hat{x}$ ” in the section “Brownian simulations” by “ $v_f = \gamma h_s \hat{x}$ ”, since the position of the bacterium along y is fixed in the simulations.
- In Bianchi 2018 PRX, the authors found the equilibrium angle of bacteria swimming along a wall without flow to be non-zero, with a mean value of 10 degrees with peak values of 30 degrees. The authors mention that non-zero values for these parameters do not change their results qualitatively, but could they provide evidence of this statement, especially for the larger values observed experimentally?

Reviewer #2 (Remarks to the Author):

This manuscript presents a theoretical, computational, and experimental study of E. coli swimming under shear flow. The authors identify a novel “oscillatory rheotaxis” motion above a critical shear rate in experimental trajectories. Using an analytical model that captures five effects of flow-wall-flagellated swimmer interactions, the authors recover various phenomena observed in swimming E. coli cells. Using Brownian dynamics simulations, the authors generate trajectories of the swimmers and identify four swimming regimes, which are confirmed against experimental trajectories.

The major claim of this manuscript is the identification and theoretical model for a novel motility mode, oscillatory rheotaxis. The theoretical model, in which various hydrodynamic + surface effects identified in other studies are unified into a single model, is carefully explained and well justified using literature values. Notably, the oscillatory frequency identified in the model is in quantitative

agreement with the experimental values at various shear rates. This is a well-integrated study with novel phenomena and conclusions that is rigorously and carefully performed and reported.

Major comments:

1. What are the critical shear rates for the observed transitions obtained in experiments? The authors state on page 7 that “The critical shear rates predicted from both numerical and analytical findings are in reasonable agreement with those observed experimentally” but this comparison is difficult to make without an explicit statement of the experimental transition shear rates. Are these transitions sharp in experiments?

2. Although the model is rigorously derived and justified, the analysis of errors statistical uncertainty, and robustness is not presented at the same high level.

(a) Please state the angular error estimated from the ellipse fitting algorithm used to determine cell orientation in the experiments.

(b) The authors nicely note on page 5 that “the results are qualitatively robust for changes in [the model] parameters” but it would be useful to add to the Materials and Methods the ranges over which robustness was tested/determined.

(c) The authors state in the Materials and Methods (p.9) that at least 100 trajectories were analyzed for each shear rate. What is the spread on the data resulting from the analysis of trajectories?

3. Is oscillatory rheotaxis likely to be significant for wild-type organisms? The authors do show in the SI that the the orientation distributions are preserved in the presence of tumbling (through simulations) but it is not clear that in a WT (non-smooth-swimming) strain the duration of straight-swimming is sufficiently long to enable such observations. This question is important for determining whether, for example, “upstream swimming in cylindrical pipes could be deterred with a right-handed patterning” (p. 8) — if a cell can tumble then this barrier may be significantly less effective.

Minor comments:

1. Caption to Figure 3: Please make sure that the caption correctly describes the layout of the figure.

2. Figure 5 caption: “flagella into to wall” is grammatically incorrect.

3. Figure 4 label/caption: the dark-blue curve in (a) is stated the text to be in the absence of shear but is labeled as “ 1 s^{-1} ” in the figure itself. Please clarify.

4. p. 8 “Langrangian”

Reviewer #3 (Remarks to the Author):

Authors characterize the motion of bacteria in a microfluidic channel, focusing on those bacteria that are swimming close to the bottom wall of the channel.

Authors provide experimental evidences of the new dynamics that they find theoretically.

The theoretical analysis is carefully carried out but it seems to be just "juxtaposed" to the experimental data.

More in detail:

1- The theoretical model is based on a far-field approach (not mentioned) that apparently is enough to qualitative capture the dynamics. Authors may comment on the limitations of this approach and when they expect larger discrepancies with experiment.

For example, which term Authors consider most relevant in the mismatch between the theoretical predictions and the experimental results in Fig.1.e.

2- Why the experimental data shown in Fig.1.d, for which the orientation is an available observable, have not been compared to Fig.4.b.?

3- Authors do not clarify why there is a left-right symmetry breaking along the z-axis such that bacteria swim to the "right". Are authors looking just a half of the channel (say the left part)?

4- Authors define the shear rate τ at the bottom walls without mentioning which are the underlying assumptions

5- Authors compare "simulations" with "theoretical" predictions but, in practice, the simulations are the numerical evolution of the theoretical model. I think that this is at the basis of the good agreement between the bottom panels of Fig.2 (theoretical model) and Fig.3 (simulations) or in Fig.3. So it is not clear why they may NOT match. Is it a matter of showing that thermal fluctuations are not very relevant? If this is so Authors should discuss why it is this so relevant. Since the Peclet number of these bacteria is quite above unity, therefore one would not expect (a priori) thermal fluctuations to be relevant.

Minor:

A- Authors should specify the meaning of the different shadows of blue in Fig.2f-h

B- The caption of Fig.3 is misleading since there are not "upper" and "lower" panels, rather "left" and "right" panels

C- I do not find fig.4.a very clear. Why not a simple 2D plot with color-coded lines (instead of histograms) such that one can appreciate the profiles "behind" the peaks of the first curves?

D- Authors define the shear rate at the bottom walls without mentioning which are the underlying assumptions. For example, this value depends on the position along the z-axis. Authors should comment on the choice they make to come up with such a prediction.

Due to the above mentioned reasons I do not recommend for publication in the present form.

Reviewer #1

Oscillatory surface rheotaxis of swimming E. coli bacteria

In this paper, the authors observe experimentally a range of rheotactic behaviors for E. coli bacteria swimming next to a wall at different shear rates, using a novel Lagrangian-tracking technique allowing them to acquire long trajectories. The experimental observations are explained theoretically by combining previously known reorientation mechanisms of a bacterium with a new chirality-induced effect. Their theoretical approach includes simulations of individual trajectories and analytical results. The authors identify a new component of rheotaxis, that is oscillation of the orientation vector at high frequency for high shear rate. They also observe both in experiments and theory trajectories biased towards the negative vorticity on top of the established positive vorticity rheotaxis. Overall, this paper combines previously explored elements to provide a complete picture of bacteria rheotaxis in varying shear rates.

While a convincing match between the experimental and theoretical description of rheotaxis for different shear rates has been achieved, the degree of novelty remains less clear and the claim that this can be a novel approach to prevent contamination is not well substantiated. It would also be important to get clarification on some of the theoretical arguments used.

First of all, we would like to thank the referee for carefully reviewing our manuscript and for providing detailed comments and suggestions. In particular, we are glad that they find we “provide a complete picture of bacteria rheotaxis in varying shear rates”, and that “a convincing match between the experimental and theoretical description of rheotaxis for different shear rates has been achieved”.

Regarding novelty, we would like to highlight that the main result of this work is the discovery of a new bacterial behaviour, the oscillatory rheotaxis, as well as the systematic characterisation of bacterial dynamics in 4 different regimes of shear, performed for the first time in our study. We have linked the characterization of these dynamics to contamination prevention, but it should be noted that this is not the only possible application. Our findings could find other applications in a broader context, such as in rheotactic cell sorting, bacterial dynamics in biological channels, and microrobot navigation complex flow environments. We agree with the referee that this was not clear initially and have rewritten our text substantially to address this valuable point.

In the following we will respond to the points raised and changes to the manuscript are marked in blue.

Impact:

- The suggested device for preventing contamination does not seem to utilize the presence of newly observed leftward trajectories/oscillations. Rather, it is based on the known tendency of

the swimmers to move right when swimming upstream (Hill et al. 2007 in the reference list). As written, it is not clear that a novel mechanism for contamination prevention has been proposed.

We thank the referee for highlighting this important point. With regard to contamination prevention, we propose two mechanisms that could help reduce upstream swimming in biomedical settings.

The first strategy is to design a spiral surface patterning in microchannels that stops cells from moving upstream. This idea is indeed based on previous work that demonstrated that bacteria have a tendency to move in the vorticity direction, especially the work by Hill et al. (2007), Kaya & Koser (2012), Marcos et al. (2012) and others. We completely agree with the referee that credit is due here, and we now mention these references explicitly when discussing this idea.

Building on this knowledge, we propose that the helix angle (i.e. pitch) of this spiral patterning should be tuned according to the dynamics of the bacterial in-plane angle. Specifically, for a right-handed helix pattern, parametrised in Cartesian coordinates as $(x, y, z) = (a \cos t, b \sin t, b t)$, the helix angle is $\alpha = \arctan(2\pi a/b)$, with $0 \leq \alpha \leq \pi$. We then require that $\alpha + \psi > \pi/2$ or $\alpha + \psi < -\pi/2$, such that the cells are directed downstream. We have rewritten the discussion section extensively to address this concern.

The second strategy is based on reducing the residence time that bacteria spend on the surface, which could reduce upstream motion because at surfaces the counter flow is the weakest (also only at surfaces bacteria are oriented mostly upstream). We would like to exploit the idea that oscillatory rheotaxis (regime III) can reduce the residence time, because oscillations of large amplitude lead to detachment from the surface and render bacteria subject to a strong downstream advection in the bulk. One could adapt microfluidic channel designs such that the shear at the surfaces is always larger than the critical shear required for these oscillations.

We have addressed these points in detail now in the discussion section.

- To follow up on the previous point, the novel observation of oscillations in trajectories and rheotaxis toward negative vorticity happen in a regime of shear where cells are advected downstream, so when their risk of swimming upstream is absent anyway. Therefore, one could argue that the knowledge of this regime of swimming is not directly relevant for contamination prevention.

We agree with the reviewer that in the regimes of oscillatory rheotaxis (III) and left-oriented rheotaxis (IV) the bacteria are advected downstream on average. However, the knowledge of the orientation dynamics in these regimes is still directly relevant for contamination prevention, for three reasons.

First, because flow rates in biological and medical settings often vary over time. To be explicit, consider a natural flow that changes between weak and strong shear rates. During the weak

flows the bacteria can swim upstream (II) and during strong flows they are advected downstream (III-IV). To evaluate the contamination potential, it is of course crucial to understand how fast the cells can move upstream, but also how well they can fight the downstream advection, because also in regimes III-IV the bacteria on the surface are on average still oriented more up- than downstream [Fig. 5]. Moreover, the orientation dynamics will also determine whether the cells will stay at the surface, or whether they will detach and be advected with much larger bulk velocities. If the cells stay at the surface and reduce the downstream motion, they can still go upstream on average as the flow varies in time. Therefore, the resistance against downstream advection can make the difference between overall contamination or not.

Second, the shear rate in biological and medical systems will also vary in space, e.g. because a channel is wider and narrower in different places, so the shear is modified. Then, different places in the device will correspond to different rheotaxis regimes (I-IV) that the bacteria can swim into. Subsequently, the dynamics in the local regimes (III-IV) will again affect the global upstream swimming ability.

Third, we have seen that in regime (II) the fluctuations can already lead to the emergence of oscillatory dynamics. This is because the orientation dynamics will be driven stochastically from the fixed points, so the first order linearization breaks down. As a consequence, though, the oscillations can already deter upstream swimming or prevent contamination in weak flows.

Indeed, these points were not mentioned in the previous version of the manuscript, and we have added a detailed discussion on this now.

- Overall, to support a claim that this study could help design devices deterring upstream swimming in medical devices or other applications, the authors should provide a more convincing case. This should include, in particular, estimates of common shear rates in medical devices, and which regimes of rheotaxis are expected in these applications.

We thank the referee for this valuable comment. There is a broad range of devices with widely varying flow rates, depending on the function, and therefore we have aimed to characterise bacterial dynamics in regimes that also span over a large range, $\gamma \in [0, 300] \text{ s}^{-1}$.

More concretely, we consider the following biologically relevant examples:

A typical urinary Foley catheter has an inner radius of $R \sim 2 \text{ mm}$, and is subjected to a flow rate of $Q = 1.5 \text{ L/day}$ for humans (Moller et al, J Clin Investig, 1928). For a cylindrically symmetric Poiseuille flow, $v = v_{\max} (1 - r^2/R^2)$, with maximum flow $v_{\max} = 2Q/(\pi R^2)$, we find the shear rate at the surfaces, $\gamma = 4Q/(\pi R^3)$. For a catheter that amounts to $\gamma \approx 2\text{-}3 \text{ s}^{-1}$ (Kaya & Koser, Biophys J, 2012), which is in the upstream swimming regime (II). However, this shear can be a hundred times larger during a ~ 20 seconds release, well inside regime (IV).

Another example in domestic or hospital pipes of radius 2cm gives $5.3s^{-1}$ for a small faucet (2L/min) or $26.5 s^{-1}$ for a shower (10L/min) (Tourigny, J Water Resources, 2018), again in the relevant regimes. Therefore, to evaluate the contamination potential, it is important to understand the orientation dynamics for a broad spectrum of shear rates. And coming back to the previous point, there can be many situations in which the shear switches between regimes (I-II) and (III-IV) over time and space.

We agree with the referee that this was not clarified before, and therefore we have rewritten the discussion section to discuss this point and include these examples.

- Similarly, could the authors discuss how their results obtained with a specific mutant of E. coli could extend to other species of bacteria? This would broaden the impact of their work.

We thank the reviewer for this point. Our model is not specific but can be used for any flagellated swimming bacterium, or other biological or synthetic microwimmers. We have now added a new section Methods §4b that gives a physical explanation how our parameters are expected to change for different swimmer geometries, and how this will affect the resulting orientation dynamics. We have also clarified in the main text how geometry plays an important role and how different bacterial species are expected to behave.

Modelling:

- The model has about 10 parameters. While the match with the experimental data is valuable, some of the parameter values were chosen quite arbitrarily. To avoid over-fitting, can the authors provide evidence that the model predictions are robust against variation in the parameters, for example (but not only) θ_0 ?

We thank the referee for this important comment. It is true that our model involves around 10 parameters, which is important to generalise our results to different microbial species. We have carefully tuned our model to experimental E. coli, so all parameters have been chosen only after extensive consideration, as described in the Methods §4a. As a result, most parameters are directly taken from measurables, and only a few remain that we justify with careful theoretical arguments.

Nonetheless, we agree that we have not demonstrated explicitly that our model is robust against changes in these parameters. Therefore, we have now performed extensive new simulations to demonstrate this, and we have adapted all our figures to show our new results.

In particular, we have now changed the pitch angle to $\vartheta_0 = -10^\circ$ to reflect the recent observations (Bianchi et al., Phys Rev X, 2017) that bacteria point towards the surface rather than swimming exactly parallel to it. This parameter change does not affect our results qualitatively, but only shifts the values of the second and third critical shear rates a little. Physically this can be

explained because a deeper pitch angle provides better anchoring for the weathervane effect, so the right and left oscillations can only start at slightly larger shear rates. Another effect is that the stability of the oscillations, and therefore the residence time at the surface, increases with a deeper pitch angle, as expected.

To demonstrate robustness against changes in the other parameters we have now included a new section, Methods §4b and a new supplementary figure S3. Here we vary each parameter over a broad range and show how the orientation dynamics and fixed points in phase space are affected.

We conclude that all four rheotaxis regimes always exist for all parameter values tested, and only quantitatively their critical shear rates are shifted up and down. For example, a large helix chirality strongly biases the motion to the right, so the third critical shear rate is increased, as expected.

In §4b we also give physical explanations for how each parameter influences the bacterial dynamics, and how different physical properties of other bacterial species affect this. We are glad that the referee brought this up and we believe it has improved our manuscript substantially.

- Could the authors clarify the derivation of Eq. 2? It appears to be the only new term in the overall orientational dynamics and its importance is stressed throughout the paper. Yet it is presented as a trial and error guess from some more complicated expression, which is not even shown. Since novelty is claimed here, more details should be given and even though Mathematica cannot simplify the long equation, it should nevertheless be displayed. In particular, since the authors write that ‘in a good approximation the helix will rotate in flow similar as a rigid rod-like particle’, isn’t Eq. 2 effectively similar to Jeffrey’s equation?

We completely agree with the referee. The derivation of equation 2 is a new result that deserves a much better presentation. We have provided a clearer description of how this solution was found in §2b of the Materials and Methods section.

Moreover, we have now provided a supplementary file “ChiralityRheotaxisExpressions.dat” that contains the full solution. We also include a supplementary Mathematica notebook file “ComparisonGraphicalFormula.nb” that shows how these expressions can be imported, and we show that they can be simplified to the expression given in the paper.

We would also like to clarify that our phrase ‘in a good approximation the helix will rotate in flow similar as a rigid rod-like particle’ was unclear. The bacterium is elongated and chiral and hence has two contributions – the Jeffery reorientation mechanism due to elongation, and, on top, the reorientation due to chirality, as written in Eq (2). This we also clarified in §2b.

- Fixed points. Section 5c on the equilibrium orientations is written in a manner that suggests that $(\psi, \vartheta) = (\pi/2, 0)$ is a fixed point or is close to some true fixed point of the equations of motion M8 and M9. The authors look for the equilibrium orientation by setting $\Omega_\psi = \Omega_\vartheta = 0$ and propose $(\pi/2, 0)$ as an approximation because ‘it’s known that bacteria swim to the right and left at high shear rates’. The resulting solution of the linearized equations is then called an equilibrium solution while M14 clearly shows that there is still a torque $[\tau = (E, 0)$ for $(\psi, \vartheta) = (\pi/2, 0)$] in the ψ direction. There is no attempt to demonstrate how close $(\psi, \vartheta) = (\pi/2, 0)$ and the 1st order correction are to the true fixed point for relevant parameter values. Are there even any other fixed points?

We thank the referee for pointing out this source of confusion, which is clarified now: We did not mean to say that $(\psi, \vartheta) = (\pi/2, 0)$ is a fixed point. Instead, we linearise the equations of motion (M10, M11) to find the fixed point on the right, at $(\pi/2 + \epsilon_\psi^R, \epsilon_\vartheta^R)$, and the fixed point on the left, at $(-\pi/2 + \epsilon_\psi^L, \epsilon_\vartheta^L)$, where we solve for the values of ϵ so that the torques do disappear, as written in equation M17. The solution to this problem is given by equation M19.

It is correct that this solution of the linearised equations of motion (M17) does not coincide exactly with the fixed points of the full equations of motion (M10, M11). However, the distance between this 1st order fixed point and the true fixed point (found numerically) is small, as shown in Fig. 5(c,d) by the dashed and solid lines for the analytical and numerical solutions, respectively. Moreover, we have also verified that there are no other stable fixed points by solving the full equations of motion numerically. Once the fixed points are approximated, one can continue with a linear stability analysis to derive the critical shear rate for oscillations (M23) and the oscillation frequency (Eq. 5).

We agree with the referee that this was not clear in the previous version of the manuscript. We have completely rewritten the section Methods §5 and the corresponding areas in the main text to clarify this, and we have also extended this text to account for the pitch angle $\vartheta_0 = -10^\circ$ that is included explicitly in the model now.

- It should be clarified that θ_0 in Eq. M16 is not the same as the equilibrium angle with no flow (Wall effects). Right now, the notation is confusing: shouldn’t the authors use ψ^* and θ^* in M16? Furthermore, for self-consistency, the authors should display θ^* in plot 4(b) or in another SI figure to check if θ^* converges to 0 as the shear vanishes, which is expected from the $\theta_0 = 0$ assumption of the ‘Wall effects’ section. It would also be interesting to verify the match between linearisation approach and numerical solution for this parameter θ^* in analogy to Fig. 4(b) for ψ_0 .

Apologies, yes this is a typo in notation that we have corrected now. The values with an asterisk are the equilibrium (fixed point) orientations and ϑ_0 is the pitch angle in the absence of shear.

Furthermore, as the referee asked and indeed for self-consistency, we have included a new panel (Fig. 5b) that displays the distribution of pitch angles as a function of applied shear, and a panel (Fig. 5d) that compares the values of ϑ^* obtained analytically from the 1st order linearization of the equations of motion (dashed lines), ϑ^* obtained numerically from the full deterministic model (solid lines), and ϑ^* obtained from the Brownian dynamics simulations, i.e. the average peak position of the orientation distribution (data points). These values are in good agreement with one another for all shear rates. Indeed, it is also verified that ϑ^* converges to ϑ_0 when the shear vanishes.

Minor comments:

- Beginning of second paragraph of the paper: “surface locomotion” could be confused with mechanism such as gliding, which are different from the flagella-based swimming above a surface described here. Reformulate.

We thank the referee for spotting this mistake. We have corrected this now.

- Second paragraph from Experimental Observation section: the explanations are unclear, because it is difficult to identify the trajectories that are mentioned on the corresponding plot. We would recommend mentioning the color of the trajectories as it is done in the Discussion section.

Yes, we agree this is a very good idea. We have implemented this now.

- Could the authors provide some idea of how frequently they observed cells swimming to the left with respect to right-swimming cells in the experiments? Right now, there no information on the experimental statistics.

We thank the referee for this important question. The events of swimming to the left are quite rare compared to swimming to the right, but they stand out quite clearly. In our simulations we observe left orientations ($\psi < 0$) at approximately 16% of the time at the largest shear rate $\gamma = 314\text{s}^{-1}$, computed from the orientation distribution in figure 5a, and about 11% at $\gamma = 46\text{s}^{-1}$. With tumbles this increases to 32% and 22% respectively, as dynamical switching is enhanced [Suppl. Fig. S2]. From the fluorescence experiments we find that around one in five bacteria move to the left at the largest measured shear rate $\gamma = 32\text{s}^{-1}$, compared to 15% in the BD simulations at the same shear rate and also with tumbles. We have discussed this in the manuscript now.

- In the experiment with fluorescently tagged flagella: the shear rate could vary by 50% between the wall and the maximum tracked position of 5 μm . Could the authors elaborate on how would such a variation of shear rate modify the dynamics?

We thank the referee for this observation. It is true that the shear rate can vary by 50% in the channels used for the fluorescently tagged flagella. We now discuss this in the text and in a new section Methods §1d. We have also added horizontal error bars in figure 1e that reflect this uncertainty.

We would also like to highlight that the main point of the experiment using bacteria with fluorescently strained flagella was to prove the existence of oscillatory rheotaxis by looking at the flagella directly, and the more accurate quantitative data for the frequency should be taken from the 3D Lagrangian tracking experiments.

- Do plots/pictures in Figs. 1 (c,d) correspond to any particular trajectory shown in Fig. 1 (b)?

Thank you. Yes, the upper panel of Figure 1c (green data) corresponds to the last trajectory in Figure 1b, which is oscillating to the left. We have now marked this track with an asterisk and clarified this in the caption. The bottom panel has not been obtained from the Lagrangian tracking technique and thus is not shown in (b).

- Panel e of Figure 1: could the authors provide the statistics behind each experimental point (how many trajectories for each point, from how many independent experiments?).

For the Fluorescence experiments, the data points at $\gamma = 3.2, 12, 20, 32 \text{ s}^{-1}$ are averages over an ensemble of $N = 7, 7, 13, 7$ trajectories, each about 5 seconds in length to ensure a good signal from the oscillations in the Fourier transform. Each trajectory results from an independent experiment.

For the 3D tracking experiments, the data points at $\gamma = 1.1, 1.9, 4.5, 9.0, 18, 49 \text{ s}^{-1}$ are averages over an ensemble of $N = 39, 18, 16, 9, 9, 3$ trajectories. These trajectories are much longer in time, around 30 seconds with a longest of 66 seconds. Therefore, we are also confident that the resulting frequencies are accurate measurements. We have included this information in Methods §1c now.

- Figure 3: Confusion in the legend, which talks of “upper panels” and “lower panels” when the figure is organised in “left” and “right” panels.

We apologise for this mistake. We have corrected this now.

- The direction of axes in Fig. 3 should match that of Fig. 1 - for example, Fig. 1 suggests that swimming upstream corresponds to swimming in the negative x-direction; the opposite is true for Fig. 3.

Thank you for pointing out this inconsistency. The axis arrows in Figure 3 were correct and consistent with Figure 1, but the frame labels were negative. We have corrected this now. We also mention explicitly now that the thick blue arrows indicate the flow direction, exactly as in Figure 1.

- In the equation for Ω^W_ψ just before paragraph “b.Flow effects”: a closing parenthesis is missing (same in equation M8?).

Thank you, we have rectified this in both instances.

- What is the method used to find the numerical solutions of the fixed points?

We use Mathematica’s built-in function FindRoot[] to obtain the fixed points, with N=1000 randomly distributed initial positions to ensure we capture all the fixed points on the left and on the right. We have verified that this gives the same result as the function NSolve[] that does not require a starting point but runs slower. We have clarified this in the methods section now.

- In Section “Methods/3D tracking experiments”: the concentrations of L-serine and PVP used in the motility buffer are missing.

Thank you, we have included this now.

- In Section “Methods/3.Simulations of surface rheotaxis”, the distance of the bacterium to the surface is fixed at value “ $\delta = W/2 = 0.5\mu\text{m}$ ”, but in “METHODS/4.Estimations of the parameters”, this distance is set to “ $h_s = 1\mu\text{m}$ ”, and in Fig. 3 caption, “ $h_s = 0.5\mu\text{m}$ ”. Which notation and value is correct? Please make notation consistent. In addition, I would suggest to replace “ $v_f = \gamma y \hat{x}$ ” in the section “Brownian simulations” by “ $v_f = \gamma h_s \hat{x}$ ”, since the position of the bacterium along y is fixed in the simulations.

We are sorry for the confusion; this was a typo. We have consistently used $h_s = 1\mu\text{m}$. We have corrected this now, and we also agree with the referee to write “ $v_f = \gamma h_s \hat{x}$ ” in the Brownian simulations section.

- In Bianchi 2018 PRX, the authors found the equilibrium angle of bacteria swimming along a wall without flow to be non-zero, with a mean value of 10 degrees with peak values of 30 degrees. The authors mention that non-zero values for these parameters do not change their results qualitatively, but could they provide evidence of this statement, especially for the larger values observed experimentally?

We thank the referee for this important question. Also see the point above. We have now included the pitch angle $\theta_0 = -10^\circ$ explicitly in our model, and changed the text and figures accordingly. Indeed, this did not change any of the dynamics qualitatively, and only shifted the critical values.

Once more, we would like to thank the reviewer for their time and detailed consideration.

Reviewer #2:

This manuscript presents a theoretical, computational, and experimental study of *E. coli* swimming under shear flow. The authors identify a novel “oscillatory rheotaxis” motion above a critical shear rate in experimental trajectories. Using an analytical model that captures five effects of flow-wall-flagellated swimmer interactions, the authors recover various phenomena observed in swimming *E. coli* cells. Using Brownian dynamics simulations, the authors generate trajectories of the swimmers and identify four swimming regimes, which are confirmed against experimental trajectories.

The major claim of this manuscript is the identification and theoretical model for a novel motility mode, oscillatory rheotaxis. The theoretical model, in which various hydrodynamic + surface effects identified in other studies are unified into a single model, is carefully explained and well justified using literature values. Notably, the oscillatory frequency identified in the model is in quantitative agreement with the experimental values at various shear rates. This is a well-integrated study with novel phenomena and conclusions that is rigorously and carefully performed and reported.

First of all, we would like to thank the referee for the comprehensive description of our work and their encouraging appraisal. Specifically, we are pleased that the reviewer finds this work “a well-integrated study with novel phenomena and conclusions that is rigorously and carefully performed and reported.”

In the following we will respond to the points raised and changes to the manuscript are marked in blue.

Major comments:

1. What are the critical shear rates for the observed transitions obtained in experiments? The authors state on page 7 that “The critical shear rates predicted from both numerical and analytical findings are in reasonable agreement with those observed experimentally” but this comparison is difficult to make without an explicit statement of the experimental transition shear rates. Are these transitions sharp in experiments?

We thank the referee for raising this important point. Even though we clearly observe all 4 regimes – and in particular the novel oscillatory and left swimming regimes – it is true that the transitions between these are not sharp, because the system is subject to intrinsic sources of fluctuations: There are natural variations in the bacterial shape and swimming speed and there is thermal and biological noise in the swimming dynamics.

These fluctuations do not only broaden the transitions, but they can also shift the critical shear rates: Above the first threshold circular swimming can coexist due to the stochastic Adler

equation. The noise also allows for oscillations to emerge below the second critical shear rate [see explanation in the last paragraph of the “analytical predictions” section].

In addition, small variations of the distance from the wall for experimental bacteria trajectories can lead to small variations of the model parameters describing the bacteria wall interactions. The dependence of the modelling outcome on such variations, and how the critical shear rates can shift in value, has now been extensively discussed in the Materials & Methods section (see also answer to point 2b below). This explains again that transitions observed in the experiments are not sharp.

Consequently, we feel it would not be entirely appropriate to extract numerical values for the experimental transition shear rates. It would probably be more accurate to say that near the thresholds these regimes coexist because of the mentioned fluctuations. Still, the critical shear rates predicted from our model do not compare unfavourably with our observations.

However, to make this more explicit, we now present a much more detailed comparison between our experiments and theory. We have performed a new analysis to not only compare the frequency of the oscillatory rheotaxis in figure 1e, but also a comparison of the equilibrium angle in figure 5c. To bring these results together in one location we have written a new section, “comparison with experiments”, to ensure a comparison of all quantities with the same level of rigour. Here we discuss (1) the swimmer dynamics itself as a function of increasing shear rate, (2) the angular dynamics, (3) the oscillation frequencies from our theory, Brownian dynamics simulations, 3D tracking experiments and fluorescence experiments, and (4) how our model might be extended to different bacterial species.

We thank the referee for highlighting this issue and we are now confident that these outcomes are presented appropriately yet thoroughly.

2. Although the model is rigorously derived and justified, the analysis of errors statistical uncertainty, and robustness is not presented at the same high level. (a) Please state the angular error estimated from the ellipse fitting algorithm used to determine cell orientation in the experiments.

We thank the referee for this important question. The uncertainty in the measurement of the in-plane orientation angle depends on the aspect ratio of the swimmer and the pixel resolution of the experiment. For the trajectory shown in Figure 1c, the length of the bacterium (with flagella included) is $a = 26 \pm 2$ pixels, and its width is $b = 7 \pm 2$ pixels. Hence, the uncertainty of the in-plane angle is $\delta\psi = \arctan[(a+\delta a)/(b+\delta b)] - \arctan[a/b] \approx 5.5$ degrees. This is indeed small compared to the observed oscillation amplitudes. Similarly, for the 3D tracking the uncertainty in the lateral swimming velocity also depends on the pixel resolution; $\delta v_z \approx v_z (\delta a/a) \approx 1.5 \mu\text{m/s}$. This is again small compared to the observed oscillation amplitudes. We have added the

corresponding error bars to the plots in figure 1c now, and we added this information in the methods section.

(b) The authors nicely note on page 5 that “the results are qualitatively robust for changes in [the model] parameters” but it would be useful to add to the Materials and Methods the ranges over which robustness was tested/determined.

We agree with the referee that the robustness in parameter space was not shown as rigorously as needed. Indeed, our model involves around 10 parameters. We have carefully tuned our model to experimental E. coli, so all parameters have been chosen only after extensive consideration, as described in the Methods §4a. As a result, most parameters are directly taken from measurables, and only a few remain that we justify with careful theoretical arguments.

In addition, to demonstrate robustness against changes in the other parameters we have now included a new section, Methods §4b and a new supplementary figure S3. Here we vary each parameter over a broad range, the biologically relevant parameter space, and show how the orientation dynamics and fixed points in phase space are affected.

We conclude that all four rheotaxis regimes always exist for all parameter values tested, and only quantitatively their critical shear rates are shifted up and down. We also give physical explanations for how each parameter influences the bacterial dynamics, and how different physical properties of other bacterial species affect this. For example, a large helix chirality strongly biases the motion to the right, so the third critical shear rate is increased, as expected.

Moreover, we have now changed the pitch angle to $\vartheta_0 = -10^\circ$ to reflect the recent observations (Bianchi et al., Phys Rev X, 2017) that bacteria point towards the surface rather than swimming exactly parallel to it. This parameter change does not affect our results, but only shifts the values of the second and third critical shear rates a little. Physically this can be explained because a deeper pitch angle provides better anchoring for the weathervane effect, so the right and left oscillations can only start at slightly larger shear rates. Another effect is that the stability of the oscillations, and therefore the residence time at the surface, increases with a deeper pitch angle, as expected.

We are glad that the referee brought this up and we believe it has improved our manuscript substantially.

(c) The authors state in the Materials and Methods (p.9) that at least 100 trajectories were analyzed for each shear rate. What is the spread on the data resulting from the analysis of trajectories?

We apologise for this confusion. The spread of this data, the standard deviation of the ensemble of frequency measurements, is shown as the vertical error bars in figure 1e. We show 2 standard deviations about the mean, which corresponds to the 95% confidence margin. It should be noted that the resulting errors bars are small in the low shear and high shear regions, where the swimmers move in circles or oscillate quickly, respectively. However, in the transition area between these regions the uncertainty is larger, as expected. We have added this information to the figure caption explicitly now.

3. Is oscillatory rheotaxis likely to be significant for wild-type organisms? The authors do show in the SI that the the orientation distributions are preserved in the presence of tumbling (through simulations) but it is not clear that in a WT (non-smooth-swimming) strain the duration of straight-swimming is sufficiently long to enable such observations. This question is important for determining whether, for example, “upstream swimming in cylindrical pipes could be deterred with a right-handed patterning” (p. 8) — if a cell can tumble then this barrier may be significantly less effective.

We are grateful for this valuable comment. Yes, we did in fact observe oscillatory rheotaxis for a WT strain (AB1157 wild-type, AD1). This is shown explicitly in figure 1d, as well as the data presented in figure 1c (bottom panel) and figure 1e (magenta stars), since all the fluorescence experiments were done with this strain. The referee is correct that tumbling does shorten the trajectory duration, but the oscillation frequencies (>1-10Hz) are typically comparable or larger than the tumbling frequency (distributed about ~1Hz), so sufficiently many oscillation periods are observed for a correct Fourier analysis.

*We also agree with the referee that tumbling could facilitate right-to-left switching, which will randomise the known tendency of bacteria to swim to the right in pipe flows. This effect would imply that the proposed right-handed patterning could be less effective. However, we find that the preferred orientation is still mostly to the right, as left swimming is relatively rare. Our simulations with tumbles show that less than 32% move to the left at the highest shear rates, and less at lower shear (Supplementary figure S2). Without tumbles less than 16% move to the left at the highest shear rates. Also experimentally we find that about one in five of the tumbling bacteria move to the left at the largest measured shear rate, $\gamma = 32s^{-1}$. Our previous experiments at higher shear rates also corroborate this (Figuroa-Morales et al., *Soft Matter*, 2015) and others (Hill et al, *Phys Rev Lett* 2007; Kaya & Koser, *Biophys J*, 2012). Therefore, it is expected that a right-handed patterning should still help in reducing contamination potential. We have clarified this in the text and added an extensive discussion on both points.*

Minor comments:

1. Caption to Figure 3: Please make sure that the caption correctly describes the layout of the figure.

Apologies, we made a mistake in the caption of figure 3 when we describing the figure layout; top and bottom panels. In the same figure was also a mistake with negative axes labels. We have corrected both now.

2. Figure 5 caption: “flagella into to wall” is grammatically incorrect.

Thank you, we have corrected this now.

3. Figure 4 label/caption: the dark-blue curve in (a) is stated the text to be in the absence of shear but is labeled as “ 1 s^{-1} ” in the figure itself. Please clarify.

We thank the referee for highlighting this confusing point. In the previous version of the manuscript we did not simulate the zero-shear case, but now we have included this benchmark explicitly. We have rectified this description and the colour codes accordingly.

4. p. 8 “Langrangian”

Indeed, we have corrected this now.

Lastly, we would like to thank the referee once more for their time and detailed consideration.

Reviewer #3:

Authors characterize the motion of bacteria in a microfluidic channel, focusing on those bacteria that are swimming close to the bottom wall of the channel. Authors provide experimental evidences of the new dynamics that they find theoretically. The theoretical analysis is carefully carried out but it seems to be just "juxtaposed" to the experimental data.

First of all, we would like to thank the referee for the thorough evaluation of our work and the detailed suggestions, comments and questions. Moreover, we are glad that the reviewer finds that "The theoretical analysis is carefully carried out" and that the "authors provide experimental evidences of the new dynamics that they find theoretically".

Regarding the "juxtaposition" of our experimental and theoretical data, we have further extended this comparison. We have performed a new analysis to not only compare the frequency of the oscillatory rheotaxis in figure 1e, but also a comparison of the bacterial orientations in figure 5c, as suggested by the referee. To bring these results together in one location we have written a new section, "comparison with experiments", to ensure a comparison of all quantities with the same level of rigour.

Below we respond to all the points raised and changes to the manuscript are marked in blue.

More in detail:

1- The theoretical model is based on a far-field approach (not mentioned) that apparently is enough to qualitative capture the dynamics. Authors may comment on the limitations of this approach and when they expect larger discrepancies with experiment. For example, which term Authors consider most relevant in the mismatch between the theoretical predictions and the experimental results in Fig.1.e.

We thank the referee for bringing up this potential source of confusion. We would like to clarify that in general we do not use a far-field approach in the model. The referee is correct that the 'wall-alignment' [Fig. 2a] was modelled using a functional form derived from far-field hydrodynamic interactions, $\Omega_W \approx -v_W \sin(2\vartheta)$ [19, 45], but a very similar double-angle sine functional form was also used to describe near-field steric interactions [20]. The prefactor (v_W) was estimated carefully, considering both hydrodynamic and steric effects, to be in agreement with these previous experimental measurements [Methods §4a].

We have now extended our model to reflect the recent observations (Bianchi et al., Phys Rev X, 2017) that bacteria point towards the surface rather than swimming exactly parallel to it. Therefore, we now use the functional form $\Omega_W = -v_W \sin(2(\vartheta - \vartheta_0))$, where $\vartheta_0 = -10^\circ$ is the pitch angle. This new model does not affect our results qualitatively, but only shifts the values of the second and third critical shear rates a little. For consistency we did not include the higher-order terms, the factor $(1+G/2(1+\cos^2\vartheta))$, which is approximately constant for small ϑ . We have now

explained more clearly that this alignment was modelled considering both steric and hydrodynamic effects.

Fig 1e shows a good qualitative agreement between experiments and the model. One important source of quantitative differences lies in the fact that in the model all parameters are constant, but these are expected to fluctuate in the experiments due to variations in bacterial shape and exact distances from the wall. We have now performed a detailed new analysis to verify the robustness of our results with respect to model variations, as described in Methods §4b and a new supplementary figure S3. Here we vary each parameter over a broad range and show how the orientation dynamics and fixed points in phase space change. We have included a detailed discussion about this in the manuscript.

2- Why the experimental data shown in Fig.1.d, for which the orientation is an available observable, have not been compared to Fig.4.b.?

We thank the referee for this valuable suggestion that we have now included. It is correct that the bacterial orientations were not compared directly, and the required data is already available. We have now extracted the peak position of the orientation distributions $PDF(\psi)$ from our experimental trajectories and plotted those in figure 5c, so that they can be compared directly to our analytical and numerical findings. As expected, at low shear rates the motion is mostly to the right. Then, at intermediate shear we see predominantly upstream orientations. At the highest shear rates the most common orientation is to the right again. Here a small fraction also moves to the left, around one in five bacteria, compared with about 15% in the BD simulations at the same shear and also with tumbling [Suppl. Fig. S2]. We have included these results in the new section, "comparison with experiments", and we have added a detailed discussion.

3- Authors do not clarify why there is a left-right symmetry breaking along the z-axis such that bacteria swim to the "right". Are authors looking just a half of the channel (say the left part)?

We would like to clarify that the left-right symmetry breaking stems from the chirality of the bacterial flagella, which leads to two effects: (1) In the absence of flow, bacteria near solid surfaces swim in the clockwise direction because the head-tail rotations. When combined with a weak shear flow, this circling motion is biased to the right, as shown in figure 3a and hence breaks the symmetry. (2) Also in the absence of surfaces there is a bias to the right because the flagellar chirality in shear flow leads to a reorientation, as described by equation 2 and shown in figure 2g. We agree with the referee that this was not explained carefully in the previous version of the manuscript and therefore we have added a detailed discussion on this point now.

Another point to clarify is that bacteria mostly move 'to the right' on one surface of the channel, but on the opposite surface the shear rate is inverted and there they mostly move 'to the left'.

4- Authors define the shear rate at the bottom walls without mentioning which are the underlying assumptions

We have defined the shear rate based on the approximation of planar Poiseuille flow,

$$\mathbf{v}_{\text{flow}} = v_{\text{max}} (1 - (2y/H)^2) \mathbf{e}_x.$$

This assumption is valid sufficiently far away from lateral walls, as it is the case in our system (at least 100micron away from side walls). The resulting shear rate at the surface where we track bacteria is then given by

$$\gamma = dv_{\text{flow}}/dy = 4v_{\text{max}}/H,$$

where H is the channel height and v_{max} is the maximum flow velocity at the channel centreline. Beyond the precision of our microfluidic pump, we have also measured this maximum flow rate with tracer particles to guarantee an accurate measurement of the shear rate.

It should also be noted that we capture bacteria within $5 \mu\text{m}$ from the surface, so the bacteria can swim a small distance away from the surface. Consequently, there is an associated uncertainty in the shear rate, especially for the fluorescence experiments in the narrower channels. We have now added error bars to address this uncertainty. We have also added a detailed discussion in the new section Methods §1d.

5- Authors compare "simulations" with "theoretical" predictions but, in practice, the simulations are the numerical evolution of the theoretical model. I think that this is at the basis of the good agreement between the bottom panels of Fig.2 (theoretical model) and Fig.3 (simulations) or in Fig.3. So it is not clear why they may NOT match. Is it a matter of showing that thermal fluctuations are not very relevant? If this is so Authors should discuss why it is this so relevant. Since the Peclet number of these bacteria is quite above unity, therefore one would not expect (a priori) thermal fluctuations to be relevant.

We thank the referee for highlighting this point. It is correct that the theoretical model and the simulations are based on the same equations of motion except for the addition of Brownian fluctuations. Whereas the Peclet number is quite above unity, there are subtle differences between the model and the simulations, since fluctuations do contribute to three main effects:

First, the fluctuations can sustain oscillations. In the deterministic model, the fixed points in regime III are stable spirals with eigenvalues that feature an imaginary component but also a small negative real component. Therefore, in the absence of noise, the cell orientations slowly converge to the stable spiral fixed points, such that oscillations are damped out. However, fluctuations about these points maintain finite oscillation amplitudes.

Second, the noise can shift the critical shear rates. Above the first threshold circular swimming can coexist due to the stochastic Adler equation, which is now discussed in the manuscript.

Third, the fluctuations facilitate dynamical switching between left and right-orientated rheotaxis [Fig. 3d], which can be envisaged as jumps in the orientation space between two local attractors.

Beyond Brownian fluctuations, most bacteria also tumble, which further contributes to these three effects. Indeed, we see more swimming to the left when we include tumble events in our BD simulations [Supplementary Fig. ~S2].

We have clarified these points in the text now, at the end of the “analytical predictions” section.

Minor:

A- Authors should specify the meaning of the different shadows of blue in Fig.2f-h

Thank you. The colours indicate the magnitude of the angular velocity. We have now included this in the legend.

B- The caption of Fig.3 is misleading since there are not "upper" and "lower" panels, rather "left" and "right" panels

Thank you, we corrected this now.

C- I do not find fig.4.a very clear. Why not a simple 2D plot with color-coded lines (instead of histograms) such that one can appreciate the profiles "behind" the peaks of the first curves?

Thank you, we have now added insets with histograms depicted as colour-coded lines, on a logarithmic scale to show the data in different ways.

D- Authors define the shear rate at the bottom walls without mentioning which are the underlying assumptions. For example, this value depends on the position along the z-axis. Authors should comment on the choice they make to come up with such a prediction.

Please see point 4 above.

Due to the above mentioned reasons I do not recommend for publication in the present form.

Finally, we would like to thank the referee once more for the helpful suggestions and the detailed comments and we hope the referee feels the new version of our manuscript can now be recommended for publication.

Reviewer #1 (Remarks to the Author):

We thank the authors for their detailed answers and appreciate the effort put in expanding the manuscript. However, based on the new version and the response of the authors we believe this work would be more suitable for a specialized journal for the following reasons:

- While we believe that the manuscript makes progress on the fundamental issue of bacterial orientational dynamics in flow by unifying the bacterial response under different shear regimes, we are not convinced that the newly described effects open the way to new applications for contamination prevention. The proposed designs rely on careful fine-tuning of shear rates or channel geometry. For example, the first suggested design of helical patterning of the channel walls not only seems to require the pitch angle to be adapted to the shear rate but also creates new issues related to the bacterial swimming near corners, which can promote upstream swimming. Also, while we agree that the oscillations in the pitch angle might promote detachment, the paper doesn't explicitly address the detachment dynamics, making the statement very speculative. Therefore, we are still not convinced that the reported results significantly impact the field of contamination prevention as suggested.
- The analytical arguments do not constitute a truly independent line of argument; rather, they only supplement the numerical simulations. While the newly added section comparing the guess solution in Eq.(2) with the full formula is convincing since machine precision is achieved, this adds value to the numerical approach, not the analytical one. Furthermore, the analysis in section 5 is minimal: the calculation in Eq. M17 is just a first step of the Newton's method – it is the first iteration of the FindRoot[] command the authors used in Mathematica anyway. The reason why this first step gives an approximate fixed point not too far from the true one is because the initial condition $(0, \pi/2)$ is informed by previous knowledge. In the future, we suggest changing the statements in the abstract ('full theoretical analysis') and in the introduction ('thorough theoretical analysis') to weaker statements such as 'full numerical analysis supplemented with analytical arguments' or something similar. Otherwise, the reader gets the impression that the three methods used in the paper: experimental, numerical and analytical are of equal importance.

Other major comments:

- The new panels in Fig. 5(b,d) indicate a positive (mean? peak?) pitch angle θ for high shear rate in simulations. This raises the question of the self-consistency of integrating the bacterial dynamics without resolving the dynamics in the y -direction; cells pointing away from the wall will likely rapidly move away from it.
- We have noticed that the equations of motion have changed between the two versions of the manuscript. While we understand some of the changes were made in response to the issues raised by referee 3, we are confused so as to why terms proportional to the geometric factor G are now neglected, more precisely, in the 'wall-effects' components.

-

Also, we noticed that, even in the $G=1$ approximation, the value of certain parameters have changed. For example, v_W is now set at 4 s^{-1} (Fig. 1 caption, Methods 4). In the older version, the same prefactor was $(1+G)*v_W^{\text{old}}$ with $v_W^{\text{old}}= 3 \text{ s}^{-1}$ (Methods 4), therefore, we would have expected $v_W=6 \text{ s}^{-1}$ in the new version. At the same time, in the new panel Fig. 5c the new parameters result in a much better agreement with experimental data now displayed (the old parameters gave different curves). We wonder if there has been any additional optimization in the choice of the parameters for fitting the data, and if so, this should be stated.

Reviewer #2 (Remarks to the Author):

The authors have satisfactorily addressed the comments and questions of the three reviewers, with significant new analyses (including statistics) presented to support the conclusions of the work. The connection between experiment and simulation, the robustness of the model, and the distinction between theory and simulation have all been strengthened and expanded.

I recommend acceptance.

Reviewer #3 (Remarks to the Author):

Authors have positively addressed my critics. I recommend for publication

Reviewer #1 (Remarks to the Author):

We thank the authors for their detailed answers and appreciate the effort put in expanding the manuscript. However, based on the new version and the response of the authors we believe this work would be more suitable for a specialized journal for the following reasons:

We would like to thank the reviewer for spending so much time reviewing this article in detail. We have taken these comments into account in the new version of this manuscript, as detailed below:

- While we believe that the manuscript makes progress on the fundamental issue of bacterial orientational dynamics in flow by unifying the bacterial response under different shear regimes, we are not convinced that the newly described effects open the way to new applications for contamination prevention. The proposed designs rely on careful fine-tuning of shear rates or channel geometry. For example, the first suggested design of helical patterning of the channel walls not only seems to require the pitch angle to be adapted to the shear rate but also creates new issues related to the bacterial swimming near corners, which can promote upstream swimming. Also, while we agree that the oscillations in the pitch angle might promote detachment, the paper doesn't explicitly address the detachment dynamics, making the statement very speculative. Therefore, we are still not convinced that the reported results significantly impact the field of contamination prevention as suggested.

We thank the reviewer for their positive assessment by stating that “the manuscript makes progress on the fundamental issue of bacterial orientational dynamics in flow by unifying the bacterial response under different shear regimes”. This fundamental issue is deeply linked to contamination prevention because it is important to understand the upstream orientation dynamics of bacteria across a wide range of shear rates. Indeed, natural flow rates range across a broad spectrum in magnitude and they vary rapidly in space and time. As the referee states, this article explicitly connects these different shear regimes with these orientation dynamics.

With this knowledge one may then start thinking about strategies that could help with preventing bacterial contamination, but it should be noted that this is not the only possible application. Our findings could find other applications in a broader context, such as in rheotactic cell sorting, bacterial dynamics in biological channels or porous media, and microrobot navigation complex flow environments. We have discussed this more clearly now.

About contamination specifically, we agree that this is a hard problem and there is no easy solution. The referee states: “The proposed designs rely on careful fine-tuning of shear rates or channel geometry.” This is an interesting challenge because, after all, bacteria also continuously adapt and fine-tune their dynamics over time. We propose a new design that could help against contamination with a helical surface patterning, based on the observed in-plane angle dynamics. We do not completely understand the referee's comment why this relies on the pitch angle, but we do agree with the referee that this

patterning should not involve sharp corners that promote upstream swimming, and we have clarified this now in the text.

The second strategy that we propose is based on the discovery of an oscillatory type of bacterial dynamics (regime III). We are glad that the referee agrees that these “oscillations in the pitch angle might promote detachment”. We completely agree that future studies should investigate detachment dynamics experimentally. Nonetheless, we do not believe that this minor point should prevent publication of our results and ideas.

- The analytical arguments do not constitute a truly independent line of argument; rather, they only supplement the numerical simulations. While the newly added section comparing the guess solution in Eq.(2) with the full formula is convincing since machine precision is achieved, this adds value to the numerical approach, not the analytical one. Furthermore, the analysis in section 5 is minimal: the calculation in Eq. M17 is just a first step of the Newton’s method – it is the first iteration of the FindRoot[] command the authors used in Mathematica anyway. The reason why this first step gives an approximate fixed point not too far from the true one is because the initial condition $(0, \pi/2)$ is informed by previous knowledge. In the future, we suggest changing the statements in the abstract (‘full theoretical analysis’) and in the introduction (‘thorough theoretical analysis’) to weaker statements such as ‘full numerical analysis supplemented with analytical arguments’ or something similar. Otherwise, the reader gets the impression that the three methods used in the paper: experimental, numerical and analytical are of equal importance.

We do not agree with the referee that our theoretical analysis is of lower importance than the simulations. Indeed, it establishes a framework for understanding surface rheotaxis, and in particular the newly discovered oscillatory motion. The theory gives predictions for the oscillation frequency, the critical shear rates, and the orientation angles themselves, in agreement with the simulations and experiments. Indeed, rather than standing alone independently, they complement each other.

It is correct that the prediction of the orientation angles (Eq. M17) is a first-order estimate based on an initial condition informed by experimental observations. However, we do not see why this reduces its value. It agrees closely with the exact numerical solution and experiments, and the resulting analysis of the oscillation frequency also agrees with the simulations and experiments. Ultimately, however, the true value of theory is that it gives clear physical insights into this highly nontrivial dynamical system, as discussed in the text and the methods sections.

Other major comments:

- The new panels in Fig. 5(b,d) indicate a positive (mean? peak?) pitch angle θ for high shear rate in simulations. This raises the question of the self-consistency of integrating the bacterial dynamics without resolving the dynamics in the y -direction; cells pointing away from the wall will likely rapidly move away from it.

We thank the referee for highlighting this potential point of confusion. Our model allows for the possibility of hydrodynamic surface attraction (Berke et al. Phys. Rev. Lett.

101, 038102, 2008) by choosing a slightly positive escape angle θ_E , such that bacteria remain hydrodynamically trapped even when they are slightly oriented away from the surface, or oriented parallel to a slightly convex surface. Only when bacteria reach the escape angle, $\theta > \theta_E$, they can leave the surface. This angle θ_E does not influence the dynamics in the model per se, but rather defines how long a bacterium stays at a surface. The escape angle can also be set to zero to remove this possibility of hydrodynamic trapping.

Our model is still self-consistent for the vertical dynamics because all the contributions from wall effects, flow effects and the coupling terms have appropriate dependences on the pitch angle θ . We note that most previous models only considered dynamics of the in-plane angle ψ , so this is already an advancement. And indeed, the resulting model predictions agree with our experiments.

- We have noticed that the equations of motion have changed between the two versions of the manuscript. While we understand some of the changes were made in response to the issues raised by referee 3, we are confused so as to why terms proportional to the geometric factor G are now neglected, more precisely, in the 'wall-effects' components.

It is corrected that we have improved our model by including θ_0 , a finite zero-shear pitch angle, as suggested by referees 1 and 3, and in accordance with the recent observations by Bianchi et al. We did not include the higher-order terms in the wall effects, the factor $\left(1 + \frac{G}{2}(1 + \cos 2\theta)\right)$, because it is approximately constant for small θ . This new model does not affect our results qualitatively, but only shifts the values of the second and third critical shear rates slightly, as explained in detail in Methods §8.

- Also, we noticed that, even in the $G=1$ approximation, the value of certain parameters have changed. For example, v_W is now set at 4 s^{-1} (Fig. 1 caption, Methods 4). In the older version, the same prefactor was $(1+G)*v_W^{\text{old}}$ with $v_W^{\text{old}} = 3 \text{ s}^{-1}$ (Methods 4), therefore, we would have expected $v_W = 6 \text{ s}^{-1}$ in the new version. At the same time, in the new panel Fig. 5c the new parameters result in a much better agreement with experimental data now displayed (the old parameters gave different curves). We wonder if there has been any additional optimization in the choice of the parameters for fitting the data, and if so, this should be stated.

Yes, because of the changed model we have performed an additional optimization in the choice of the parameters. We have carefully explained this choice of parameters in Methods §7.

Once more, we would like to thank the reviewer for their time and helpful comments.

Reviewer #2 (Remarks to the Author):

The authors have satisfactorily addressed the comments and questions of the three reviewers, with significant new analyses (including statistics) presented to support the conclusions of the work. The connection between experiment and simulation, the robustness of the model, and the distinction between theory and simulation have all been strengthened and expanded.

I recommend acceptance.

We are grateful for this positive recommendation.

Once more, we thank the reviewer for their time.

Reviewer #3 (Remarks to the Author):

Authors have positively addressed my critics. I recommend for publication

We are grateful for this positive recommendation.

Once more, we thank the reviewer for their time.